# A single early-in-life macrolide course has lasting effects on murine microbial network topology and immunity

Victoria E. Ruiz[1], Thomas Battaglia[1], Zachary D. Kurtz[1], Luc Bijnens [2], Amy Ou [1], Isak Engstrand[3], Xuhui Zheng[1], Tadasu Iizumi[1], Briana J. Mullins [1], Christian L. Müller [4], Ken Cadwell [5], Richard Bonneau[4,6,7], Guillermo I. Perez-Perez[1] & Martin J. Blaser [1,8]

Broad-spectrum antibiotics are frequently prescribed to children. Early childhood represents a dynamic period for the intestinal microbial ecosystem, which is readily shaped by environmental cues; antibiotic-induced disruption of this sensitive community may have long-lasting host consequences. Here we demonstrate that a single pulsed macrolide antibiotic treatment (PAT) course early in life is sufficient to lead to durable alterations to the murine intestinal microbiota, ileal gene expression, specific intestinal T-cell populations, and secretory IgA expression. A PAT-perturbed microbial community is necessary for host effects and sufficient to transfer delayed secretory IgA expression. Additionally, early-life antibiotic exposure has lasting and transferable effects on microbial community network topology. Our results indicate that a single early-life macrolide course can alter the microbiota and modulate host immune phenotypes that persist long after exposure has ceased.

[1] Departments of Medicine and Microbiology, New York University School of Medicine (NYUSM), New York, NY 10016, USA. [2] Janssen R&D, Janssen Pharmaceutical Companies of J&J, Turnhoutseweg 30, Beerse 2340, Belgium. [3] Department of Molecular Medicine and Surgery, Karolinska Institutet, SE-171 77 Stockholm, Sweden. [4] Center for Computational Biology, Flatiron Institute, Simons Foundation, New York, NY 10010, USA. [5] Kimmel Center for Biology and Medicine at the Skirball Institute, NYUSM, New York, NY 10016, USA. [6] Department of Biology, Center for Genomics and Systems Biology, NYU, New York, NY 10003, USA. [7] Courant Institute of Mathematical Sciences, NYU, New York, NY 10012, USA. [8] New York Harbor Department of Veterans Affairs Medical Center, New York, NY 10010, USA. Correspondence and requests for materials should be addressed to M.J.B. (email: Martin.Blaser@nyumc.org)

Antibiotic use in clinical medicine is excessive; > 250 million antibiotic courses were prescribed in the USA in 2010[1], with ~50 million prescribed to children[2]. Antibiotic prescription rates are highest in the first 2 years of life with broad-spectrum β-lactams and macrolides most frequently prescribed for upper respiratory tract infections[3].

Clinical and epidemiologic studies have associated early-in-life antibiotic exposures with an increased risk of asthma, allergies and inflammatory bowel disease[4–6]. The autochthonous microbiota co-evolved with their hosts over millions of years developing mutualistic relationships with its host[7]. Such relationships dictate host physiological processes including

epithelial barrier function, nutrient metabolism, mucosal immune activation and protection[8], one hypothesis is that an antibiotic-altered microbiota may induce such pathologies[9].

The period between birth and 3 years of age is critical for the development of the intestinal microbiota[10]. Factors such as delivery mode, diet and antibiotics have considerable effects on the stability, succession and resilience of the intestinal microbial community[11]. With doses relatively mirroring the pharmacokinetics used in treating human infections, pulsed antibiotic treatment (PAT) induced substantial changes in murine metabolic development with macrolide antibiotics exhibiting stronger effects[12]. The differential impact of macrolide vs β-lactam antibiotic classes on early-life microbial communities and host health was further highlighted in a recent clinical study[13].

Prior studies have used massive antibiotic exposures to perturb immunological development in mice[14, 15]. Here we examine the role of one macrolide PAT course on intestinal microbial community dynamics and network structure and on the host's developing immune system. We show that even a single antibiotic course, given early in life, leads to profound and long-lasting immunological changes in mice, and that an altered microbiota with altered keystone taxa is both necessary and sufficient to explain these effects.

## Results

**Effect of macrolide courses on the intestinal microbiota**. To determine if a single antibiotic course introduced early in life is sufficient to lead to durable changes in both the microbiota and in the host, we compared the effect of exposing mice to a single pulsed antibiotic course (PAT1) vs a 3-course (PAT3) regimen (Fig. 1a). The first course was administered to both PAT1 and PAT3 pups while nursing at postnatal day 5 (P5) for 5 days. The PAT3 group received two additional courses at P27 and P36 for 3 days (Fig. 1a). Since the first antibiotic course was given to the dams during nursing, we sampled both pups and their mothers (dams) to compare the effects of the course on developing (pups; P5–10) and mature (dams; ~12 weeks old) microbiota.

Exposure to PAT1 and PAT3 regimens early in life altered intestinal microbial community composition and dynamics. After PAT1, both phylogenetic diversity and community evenness were decreased, persisting for >6 weeks, with more extensive changes in the PAT3 mice (Fig. 1b, Supplementary Fig. 1a and Supplementary Table 1). After weaning, the microbial communities in the PAT1 and PAT3 groups are similar until the additional antibiotic courses in the PAT3 group (Fig. 1c). However, 2 weeks after the final antibiotic pulse, the PAT3 microbial communities reverted to the PAT1 community state (Fig. 1c). Inter-group community variation was higher in both the PAT1 and PAT3 groups compared to control, indicating that the greatest effect was from the first antibiotic course (Fig. 1d, Supplementary Fig. 1b and Supplementary Table 2). At P52,

13 days after the final PAT3 exposure, bacterial load was decreased compared to control, but the PAT1 group had recovered (Supplementary Fig. 1c). Using either one or three antibiotic courses decreased relative abundances in the Bacteroides family S24-7, genus *Bifidobacterium*, and segmented filamentous bacteria (SFB), and increased relative abundances in Enterobacteriaceae and *Akkermansia* (Fig. 1e and Supplementary Fig. 1d–e). In contrast to pups, adult females, ~12 weeks of age, exposed to one antibiotic pulse showed intermediate effects on microbial phylogenetic diversity and community composition (Fig. 1b, c). The adult mice showed improved microbial community recovery compared to the antibiotic-exposed pups, shown by a decrease in inter-group UniFrac distance in PAT vs control (Fig. 1d). As in the pups, by sacrifice there was no effect on the total bacterial load 50 days after the PAT1 exposure in the dams (Supplementary Fig. 1c). Taken together, these findings indicate that a single antibiotic course, early in the mouse life, is sufficient to lead to long-term alterations on intestinal microbial communities, greater than in adult mice.

**Effects of macrolide courses on host immune features**. Since our prior studies of PAT showed substantial changes in gene expression in the ileum[16], consistent with predictions based on the role of the microbiota in conventional vs germ-free animals[17], we first assessed the effect of the number of antibiotic courses on ileal gene expression. Unsupervised hierarchical clustering of differential ileal genes revealed strong antibiotic exposure signatures, with all of the PAT mice segregating from the controls (Fig. 2a and Supplementary Data 1). Compared with controls at P52, after FDR correction, 148 genes were differentially expressed in either of the two PAT groups; 137 (93%) were in the PAT3 mice, with 63 shared with PAT1 and 10 unique to the PAT1 mice (Fig. 2a). Thus, although the greatest effects occurred in the PAT3 mice, even a single PAT course affected gene expression, continuing > 40 days after the last exposure.

Since a single PAT treatment showed significant decreases in genes involved in immunity, we asked whether the antibiotic exposure altered local and peripheral T-helper populations. Total numbers of small intestinal lamina propria (SI-LP) and splenic leukocytes were not significantly different between the control, PAT1 and PAT3 pups ($p > 0.05$, Kruskal–Wallis non-parametric test used with Dunnett's multiple comparisons test) (Supplementary Fig. 1g). However, intestinal $CD4^+$ $IL17A^+$ lymphocytes were decreased in both pup groups, with no differences in the dams (Fig. 2b), Th1 ($CD4^+$ $IFN\gamma^+$)-expressing cells were not affected but in the PAT3 group, $Foxp3^+$ regulatory T cells were increased (Supplementary Fig. 1h). Splenic $CD8^+$ lymphocytes were reduced in both pup groups, while $CD4^+$ cells were decreased after PAT3 exposure (Fig. 2c).

Due to known roles of the microbiome and Th17 lymphocytes on IgA expression, we next asked whether the single antibiotic course affected mucosal and systemic IgA levels. In the pups, at P27, after both PAT groups had received only a single course,

**Fig. 1** Effect of number and timing of antibiotic doses on intestinal microbial communities. **a** Study design: 5-day-old C57BL/6 pups were treated with one course of tylosin at P5 for 5 days (PAT1 group) through their mother's milk, or with two additional doses at P27 and P36 for 3 days each (PAT3 group). Twelve-week-old dams were treated with one course at pup P5. Sample sizes for dams were n = 7 (PAT), n = 3 (control). Sample sizes for their pups were n = 12 (control), n = 17 (PAT1), n = 19 (PAT3). **b** Mean (±SEM) unrarified α-diversity using the phylogenetic diversity (PD) metric in fecal samples of dams and offspring (pups) over the course of the experiment. *Solid lines* represent female pups n = 3 (control), n = 8 (PAT1), n = 9 (PAT3) and dams; *dotted lines* are male pups; n = 9 (control), n = 9 (PAT1), n = 10 (PAT3). Statistical analysis performed using two-sample *t*-test with Monte Carlo permutations, for statistical significance, see Supplementary Table 1. **c** Unweighted UniFrac analysis of fecal specimens of pups and dams visualized by principal coordinate analysis (PCoA). The three components explain 42.3% of the total variance. **d** Intergroup unweighted UniFrac distances averaged over independently drawn sample pairs (subsampled without replacement and replicated 999 times) for the time points and groups shown in (**c**) shading indicates initial period of antibiotic exposure (for statistical significance, see Supplementary Table 2). **e** Mean relative abundance of taxa in control, PAT1 and PAT3 groups over the course of the experiment in fecal (over time), cecal and ileal samples

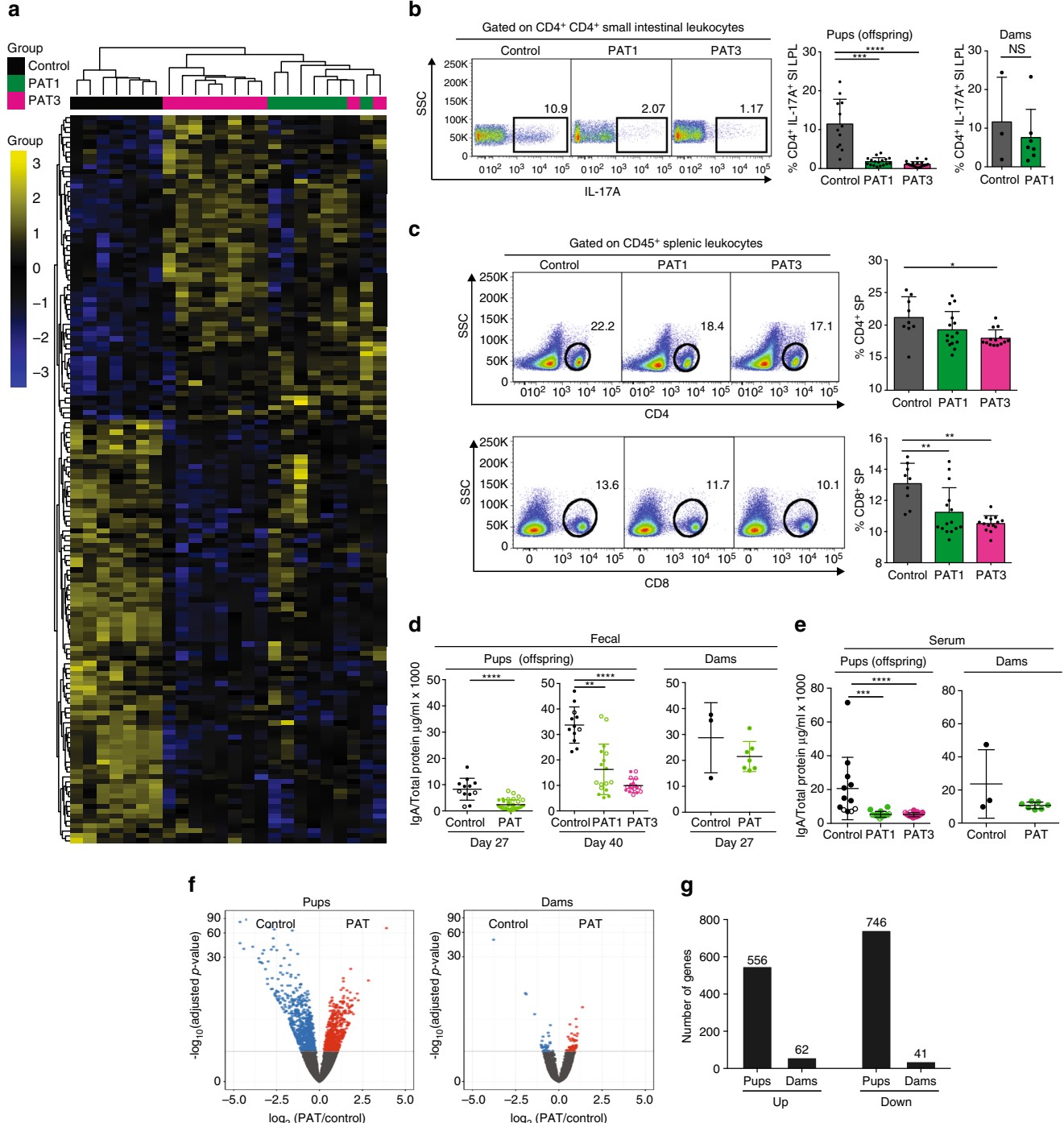

**Fig. 2** Administration of one or three macrolide courses alters ileal gene signatures and T-cell populations. **a** Heatmap with unsupervised clustering of FDR-corrected, differential ileal gene expression profiles using Nanostring technology (Control ($n = 7$), PAT1 ($n = 8$) and PAT3 ($n = 9$)). After one-way ANOVA, Tukey HSD multiple comparison testing and FDR-correction, 148 genes remained significantly different between the groups. **b** Mean (±SD) frequency of small intestine lamina propria lymphocytes after PAT1, PAT3 or control exposure. Cells were isolated at sacrifice from pups (mean P52) and their respective dams (P61). Populations were gated on live CD45⁺CD4⁺ cells, and representative frequencies of CD4⁺IL17A⁺ cells shown. **c** Frequency of splenic lymphocytes after PAT1, PAT3 or control exposure. Cells were isolated from pups at sacrifice, gated on CD4⁺ and CD8⁺ populations, and representative frequencies shown. Data compiled from three experiments ($n = 9$–19 mice/group). **d** Secretory IgA (µg/ml) (mean ± SD) in fecal samples of control and PAT-exposed dams at P27 and their pups at P27 and P40. **e** Serum IgA (µg/ml) (mean ± SD) in control and PAT-exposed mice in pup and dam samples at mean P52 or P61, respectively. For panels (**b**–**e**) significance testing performed using the Mann–Whitney non-parametric test or the Kruskal–Wallis non-parametric test used with Dunnett's multiple comparisons test. **f** Volcano plots showing significant global ileal transcriptomic alterations in pups and dams, determined by RNAseq; $n > 16,000$ genes examined, control ($n = 3$) and PAT ($n = 6$) dams, respectively; control ($n = 3$) or PAT ($n = 4$) pups. **g** Differentially expressed genes in relation to PAT exposure in P52 pups, and their dams at P61, determined by RNAseq, and depicted by direction of differences. For all panels: *$p < 0.05$, **$p < 0.01$, ***$p < 0.001$, ****$p < 0.0001$

fecal sIgA levels were lower than in the control mice ($p < 0.001$; Kruskal–Wallis non-parametric test with Dunnett's multiple comparisons test) (Fig. 2d). By P40, sIgA levels rose in the controls as expected, reflecting increased PIGR-mediated active transport[18], but was lower in the PAT1 group, and more so in the PAT3 mice; however, this phenomenon was absent in the PAT-dams (Fig. 2d). We also assessed fecal sIgA levels in the control and PAT dams before treatment to determine if there was variability in sIgA expression that could have mitigated the PAT-induced sIgA effects in the dams and her subsequent offspring. Fecal samples collected before breeding and during gestation showed no differences in fecal sIgA in dams that later were randomized into control and PAT groups (Supplementary Table 3). At sacrifice, IgA levels in serum were decreased in both the PAT1 and PAT3 pups, but not in their PAT1-exposed dams (Fig. 2e).

Since the dams of the PAT1 pups were exposed at the same time to the single antibiotic pulse but were adults rather than neonates (Fig. 1a), we examined whether host age at antibiotic exposure affected gene expression. Using RNAseq to compare global ileal gene expression between control and PAT-exposed animals, we found fewer differentially expressed genes in the dams than in the parallel comparisons of the pups (Fig. 2f, g). In total, only 47 of the 103 genes that were different between PAT and control mice were shared by the PAT-exposed dams and pups. Using Ingenuity Pathway Analysis, we examined the differentiating pathways that were shared between pups and dams. Of the 12 most significant gene expression pathways differentiating the PAT and control pups, most affected immune processes and were down-regulated. However, in the PAT dams, essentially all significant pathways were up-regulated genes and related to metabolic processes (Supplementary Table 4).

In total, a single antibiotic regimen was sufficient to alter mucosal and systemic immunological markers in the pups; comparisons with their similarly exposed dams provide evidence that the host effects depended on the age at exposure.

**Microbiota presence is necessary for PAT effects**. In addition to their anti-bacterial activity, macrolides are also known for immunomodulatory properties[19, 20]. To determine if the observed immunologic effects were due to the direct effects of the antibiotic on the host, or due to antibiotic-induced changes in the microbiome, we exposed germ-free (GF) mice with the same P5-10 tylosin pulse (PAT) or not (control), and followed mice until sacrifice at P50 (Fig. 3a); in parallel, we exposed SPF mice to PAT or not. As we now expected in the PAT-exposed SPF mice, microbial diversity, community structure, and composition were altered throughout the experimental course (Fig. 3b, c).

Cecal size was increased in GF mice, as expected, and trended larger in the PAT-exposed SPF mice (Fig. 3d). At sacrifice, cecal sIgA levels in the GF-control mice were lower than the SPF-control group, as expected, confirming the important role of the microbiota in priming the sIgA response[21]; sIgA levels in the PAT-exposed and unexposed GF mice were not different (Fig. 3e), indicating lack of direct antibiotic effect.

To assess potential PAT effects on immunologic development, we examined ileal gene expression in the four study groups at P50. Between all groups, 145 genes examined were different. Between the (antibiotic-free) control SPF and control GF mice, 141 (26%) of the 547 genes were different, reflecting the expected profound influence of the GF state on immunological development[4]. In aggregate, the gene expression profile of the antibiotic-unexposed SPF mice (normal) clustered far apart from the SPF-PAT-exposed, and the germ-free mice (Fig. 3f). By unsupervised hierarchical clustering, all GF mice, whether or not

PAT-exposed, formed a single cluster (Fig. 3g and Supplementary Data 2), indicating lack of macrolide effect in mice without microbiota. In contrast, SPF mice showed the expected strong PAT effect compared to the unexposed controls; importantly, the profiles for PAT-exposed SPF mice clustered with the GF mice, and not with the SPF controls (Fig. 3g), even 40 days after the antibiotic exposure, indicating a persistent effect. Between the control and PAT-exposed SPF mice, multiple ($n = 55$) genes had differential expression, but in the GF mice, there were no genes that were differentially expressed between PAT and control. Thus, gene expression profiling supported the substantial broad immunological differences between the GF and SPF states, the similarity of PAT-exposed SPF to GF even > 40 days after the antibiotic exposure, and the lack of PAT effect in the GF state.

We examined small intestinal lamina propria T-cell populations in the four groups of mice. As we now expected in the PAT-exposed compared to the (unexposed) SPF controls, TCRβ+, TCRβ+ CD4+ Rorγt+, and TCRβ+ CD4+ IL17A+ SI-LP lymphocytes were decreased (Fig. 3h). In the GF mice, all of these cell populations were depleted, without significant differences between PAT and untreated controls. Moreover, ileal immune gene expression, sIgA expression, and intestinal immune populations were substantially low in GF mice and were not further affected by PAT treatment. This work provides evidence that the immunological effects caused by early-life PAT in mice are not a direct antibiotic effect but an effect of an antibiotic-altered microbiota.

**A PAT-altered microbiota is sufficient to impair immune features**. Since a PAT-perturbed microbiota is necessary to impair host immunity, we next asked whether or not it is sufficient for the immunologic effects observed. We focused on changes in fecal sIgA, with the strong evidence from the prior experiments, and because we could obtain serial data from individual mice. To accomplish this, first we exposed mice to PAT or not (control) to serve as donors for microbiota transfer (Fig. 4a). On P12, we harvested cecal contents from the PAT and control mice as two separate pools, and gavaged these donor materials into P35 GF-recipient mice, conventionalizing them. For the first 6 days after transfer, all recipients had low sIgA levels (Fig. 4b). Fecal sIgA levels started rising from post-transfer day 12 and leveled off after day 25. However, the area under the curve was significantly smaller in the recipients of the PAT-perturbed microbiota (random effects ANOVA, $F_{1,10} = 43.36$, $p < 0.0001$) (Fig. 4b). At sacrifice (day 77 after transfer), the recipients of the PAT-perturbed microbiota had lower splenic TCRβ+ populations than control recipients (Fig. 4c). The differential immunologic phenotypes were independent of microbial density (Supplementary Fig. 2a), since 16S levels were similar for both recipient groups. Importantly, SFB were negligible in all samples until day 77 after transfer, so the early differential effects between the PAT and control microbiota recipients were SFB-independent (Supplementary Fig. 2b). This study provides evidence that a PAT-perturbed microbiota (regardless of SFB status) is sufficient to impair host immunological development in recipient mice, involving both humoral and cell-mediated immune characteristics.

**Dynamics of antibiotic-perturbed microbiota after transfer**. We next assessed how microbial communities re-assemble in naive (GF) recipients to detail differences due to an inoculum that had been antibiotic-perturbed in the prior host (Fig. 4d, e and Supplementary Fig. 2c). Fecal samples were collected serially after gavage to evaluate the characteristics of the microbiota after transfer and to determine whether the PAT and control

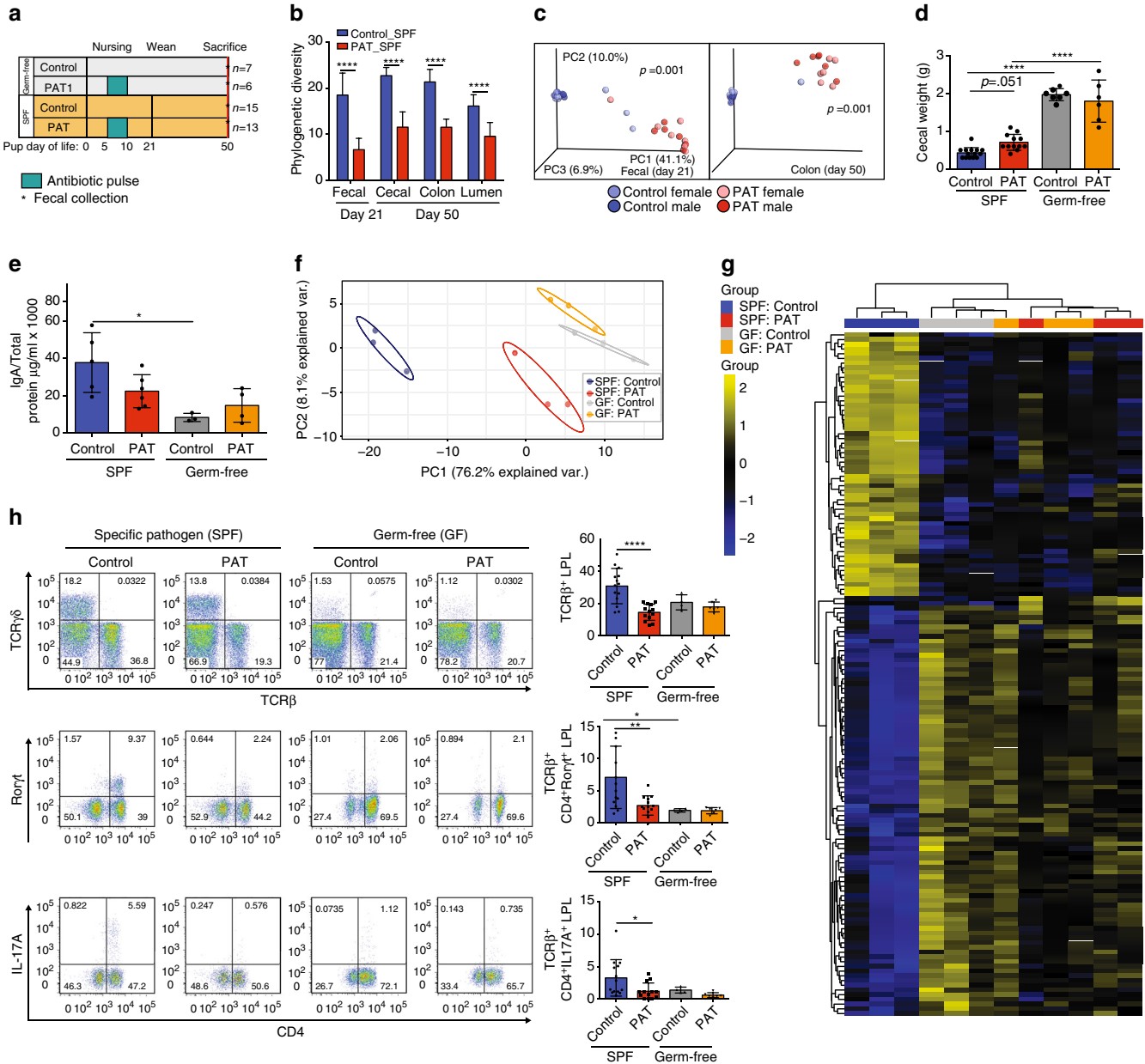

**Fig. 3** Physiologic effects of PAT in germ-free and conventional specific pathogen-free mice. **a** Study design: Germ-free (GF) and Specific pathogen-free (SPF) C57BL/6 mice were treated with one course of tylosin on P5 for 5 days; SPF mice (control ($n = 15$), PAT ($n = 13$)), GF mice (control ($n = 7$)), (PAT ($n = 6$)). All mice were sacrificed at P50. **b** Mean ($\pm$SD) unrarified $\alpha$-diversity using the phylogenetic diversity (PD) metric of fecal samples at P21 and cecal and luminal contents at P50 in control and PAT SPF mice. Statistical analysis performed using two-sample $t$-test with Monte Carlo permutations, for statistical significance, see Supplementary Table 1. **c** Unweighted UniFrac analysis of fecal specimens in male (*darker circles*) and female (*lighter circles*) PAT or control SPF mice, depicted by principal coordinate analysis (PCoA). The three components explain 58% in total variance. Statistical analysis performed using an Adonis test, $p = 0.001$. **d** Aggregate cecal size in SPF and GF mice at P50 (mean ($\pm$SD)). **e** Secretory IgA (mean ($\pm$SD)) in luminal contents of offspring at P50, SPF mice (control ($n = 5$), PAT ($n = 6$), GF mice (control ($n = 3$), PAT ($n = 4$)). **f** PCoA of differential gene expression profiles in GF and SPF Control and PAT-exposed groups, with 76.2 and 8.1 % of the variance represented in PC1 and PC2, respectively. **g** Heat map of unsupervised hierarchal clustering of differential ileal gene expression profiles. After one-way ANOVA, Tukey HSD multiple comparison testing and FDR-correction, 145 genes were differentially expressed either between control vs PAT SPF, or GF vs SPF control groups ($n = 3$/group). **h** Frequency of small intestine lamina propria lymphocyte (LPL) populations in PAT-treated GF and SPF mice (mean ($\pm$SD)). Intestinal LPLs were isolated from offspring at sacrifice. Representative flow cytometry plots were gated on live TCRβ⁺ cells, with frequencies of TCRβ⁺, IL17A⁺CD4⁺, and Rorγt⁺CD4⁺ cells shown. Data are compiled from two experiments. Statistical analysis performed using the Kruskal–Wallis non-parametric test with Dunnett's multiple comparisons test. For all panels: *$p < 0.05$, **$p < 0.01$, ****$p < 0.0001$

microbiota remain distinct. The inoculum from the PAT-exposed donors had lower phylogenetic richness and evenness than the inoculum from the controls (Fig. 4e, Supplementary Fig. 2c and Supplementary Table 1), reflecting the strong immediate PAT effects. With conventionalization, both inocula colonized the

recipients with similar efficiency, with stable total bacterial levels reached by day 12 post-transfer (Supplementary Fig. 2a); however, community $\alpha$-diversity, composition and community structure markedly differed (Fig. 4d, e and Supplementary Fig. 2d). One-day after transfer, $\alpha$-diversity in fecal samples fell

for both groups of recipients, and then progressively increased until the time of sacrifice (Fig. 4e and Supplementary Table 1). However, community evenness was lower in the PAT-conventionalized mice until day 43, and remained lower in the ileum at sacrifice (Supplementary Fig. 2c and Supplementary Table 1). The community structures of the control and PAT

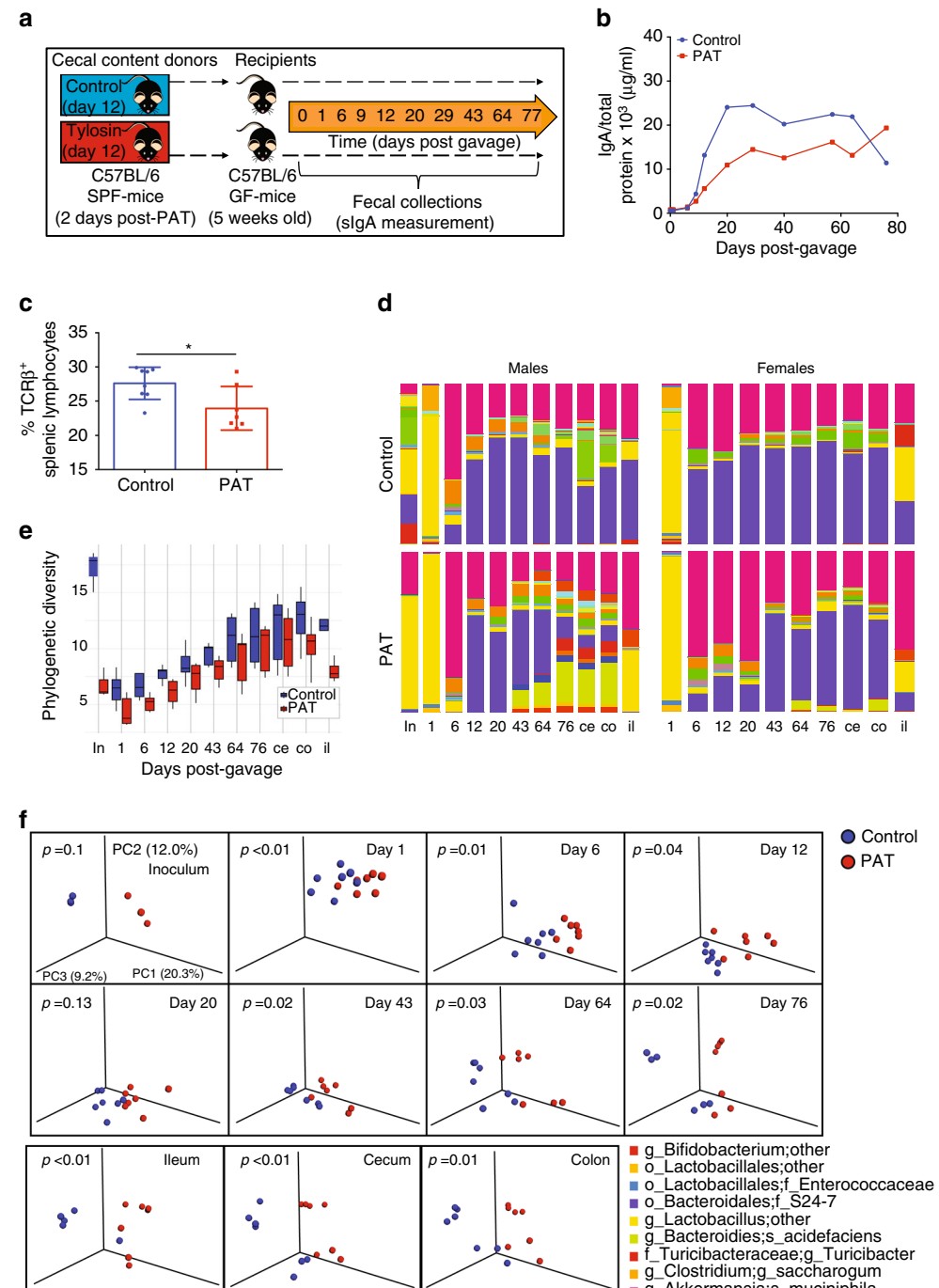

**Fig. 4** Dynamics of PAT-perturbed microbiota after transfer and effects on host immune phenotypes. **a** Study design: donor mice received non-acidified water alone (controls) or drinking water with tylosin (PAT) from P5 to P10 and were sacrificed at P12, then cecal contents were harvested and pooled. Germ-free (GF) C57BL/6 mice at P35 were gavaged with cecal contents from the P12 control or PAT-exposed donors ($n = 7$ per group). **b** Fecal secretory IgA from day 0 to 76 post-gavage in the now-conventionalized mice that had received control or PAT cecal contents. A random effects repeated measures analysis of variance (ANOVA) was used to test for differences in IgA values between the PAT and the control recipients, taking into account the correlated structure of the measurements within subjects, ($F_{1,10} = 43.36$, $p > 0.0001$). **c** Frequency of splenic TCRβ+ lymphocytes at sacrifice, 77 days post-gavage, (mean (±SD)). Statistical analysis was performed using the Mann–Whitney $U$ non-parametric test; *$p < 0.05$. **d** Relative abundance of taxa in inoculum (in), fecal, cecal (ce) and colon (co) samples over the course of the experiment. **e** Median (±IQR) unrarified microbiota α-diversity over the course of the experiment in recipients of the control or PAT-perturbed inoculum, in fecal samples, or ileal (il), cecal (ce) and colonic (co) contents at sacrifice. **f** Unweighted Unifrac analysis of inoculum and fecal specimens from groups conventionalized with control (*blue*) or PAT-perturbed (*red*) microbiota, visualized in principal coordinate analysis (PCoA); the three components explain 43% of the total variance for each panel. Statistical analysis of intergroup UniFrac distances performed by Adonis test, with $p$-values shown

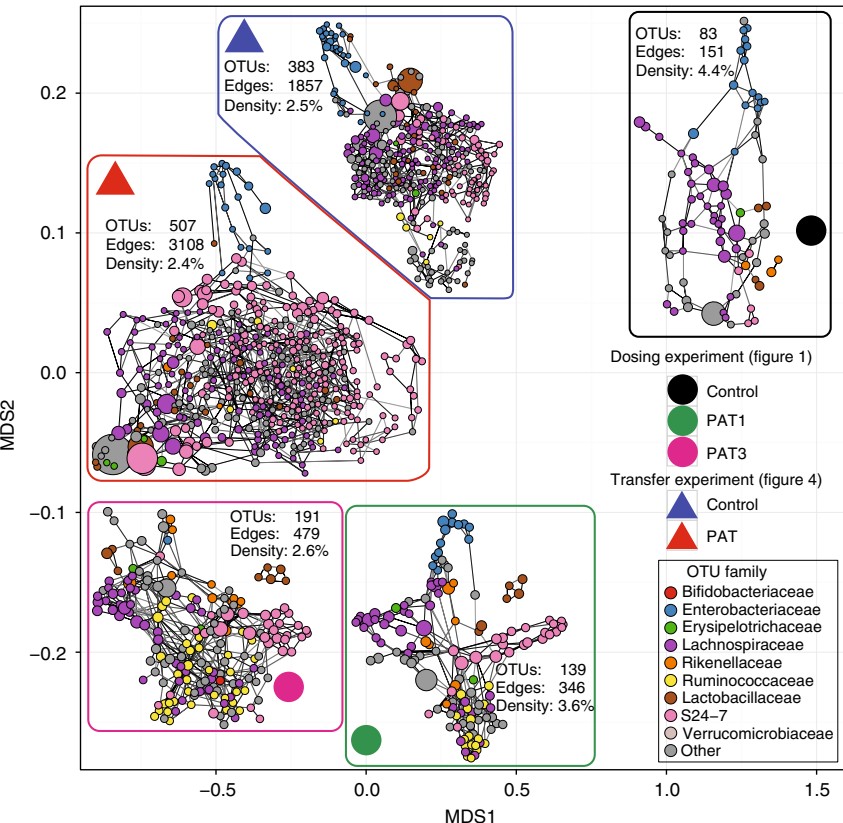

**Fig. 5** Network properties recapitulate experimental perturbation. Networks were inferred using the SPIEC-EASI pipeline. To compare graphs, a two-dimensional embedding of graphlet correlation distances (using multidimensional scaling (MDS)) was used with the network positions shown as *circles* (from the dosing experiment (see Fig. 1)) or *triangles* (from the transfer experiment (see Fig. 4)). The corresponding networks on the MDS plot were overlaid near their respective positions in the embedding, and shown in a force-directed layout. Within each graph, nodes are colored by OTU family, with size proportional to (geometric) mean abundance, and edge width proportional to confidence score, as determined by stability selection. Each panel shows the number of OTUs (identified as those present in > 20% of samples), corresponding number of edges and calculated edge density (predicted edges/number of pairs of nodes)

inocula were distinct, and fecal samples in the recipients were different from day 12 after transfer through the experiment's end (Fig. 4f). PAT-conventionalized mice exhibited sex-specific differential taxonomic profiles beginning 6 days after transfer and remaining until the end of the experiment; nonetheless, community diversity in these mice remained distinct from control. These consistent differences indicate the fidelity of the microbial compositions colonizing naïve hosts over a period of months. The initial microbiota colonizing the control inocula recipients were enriched in *Ruminococcus* and Erysipelotrichaceae species, S24-7, and *Dorea*, whereas the recipients of the PAT inocula had higher representation of *Akkermansia*, *Lactobacillus*, Rikenellaceae, and particular Clostridia species (Supplementary Fig. 2d). One day after transfer, genus *Lactobacillus* bloomed in both groups of recipients, showing their role as pioneers even in the absence of milk (Fig. 4d). Over time, in both recipient groups, Enterobacteriaceae, *Akkermansia* and family S24-7 bloomed, with greater S24-7 abundances in the controls (Fig. 4d).

**IgA-coated bacteria after control or PAT microbiota transfer.** The transfer experiment allowed exploring the interplay between the PAT-induced sIgA alterations and the microbial populations bound to sIgA. Using IgA-Seq, we identified sIgA-coated bacteria after the conventionalization with control or PAT microbiota. The community composition of the sIgA-recognized bacteria differed between the PAT and control microbiota recipients

(Supplementary Fig. 3a). *Akkermansia* was highly abundant in the sIgA+ fractions from both PAT and control samples for most of the experiment, indicating a strong sIgA affinity to *Akkermansia* if present in the microbial community, independent of exposure (Supplementary Fig. 3b). Next, using the IgA-coating index (ICI), we identified taxa highly represented in the sIgA-negative fractions, including family Lachnospiraceae, S24-7, and *Clostridium citroniae*, whereas genus *Lactobacillus* and a particular Clostridia taxon were consistently over-represented in the sIgA-positive fractions (Supplementary Fig. 3c, d). In the sIgA-positive fractions in the control inocula-recipients, there was enhanced recognition of S24-7, Dorea, Lactobacillus, Erysipelotrichaceae, Clostridial taxa and Mollicutes, but in the PAT microbiota-recipients, only *Bacteroides acidifaciens*, genus *Sutterella*, and certain Clostridia were enriched, and only late in the experiment (Supplementary Fig. 3d). In total, these results indicate that compared to control, the PAT inocula recipients have a distinct IgA repertoire associated with a markedly altered community of host-interactive bacteria.

**Associations between sIgA and microbial taxa.** To assess the relationships between particular operational taxonomic units (OTUs) and sIgA levels we applied multi-level, sparse PLS (sPLS) models. We focused on the fecal microbiota in the dosing experiment (Fig. 1), and used out-of-sample prediction of fecal microbes from the transfer experiment (Fig. 4) to drive model selection. Overall, 8 of the 53 OTUs commonly abundant in both

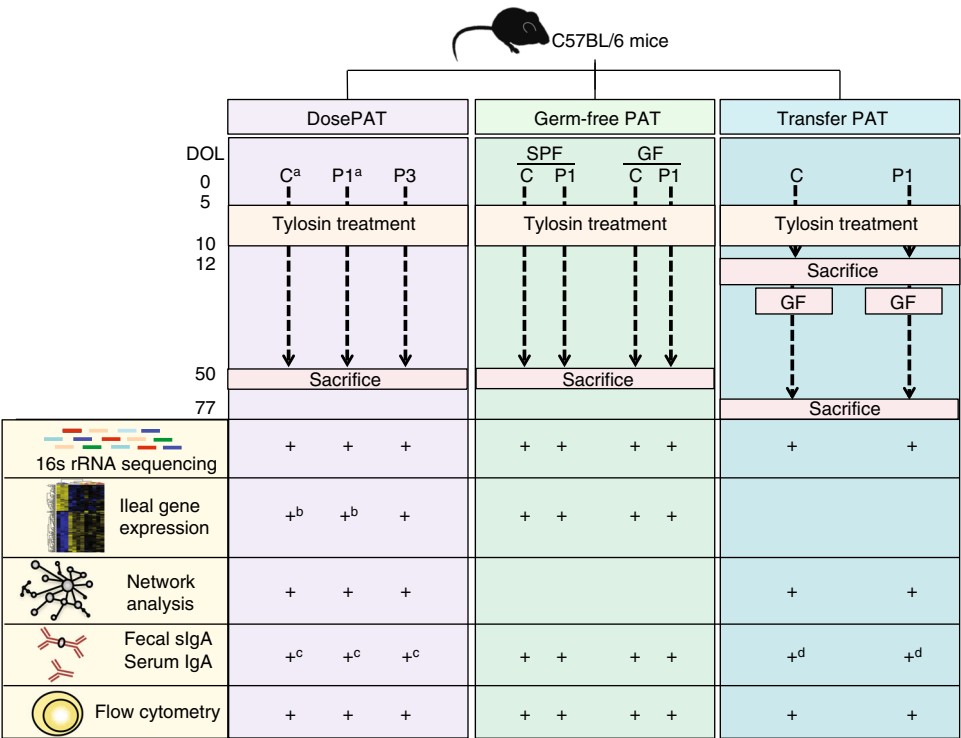

**Fig. 6** Schematic of the experiments performed in this study. Footnotes: All + refer to samples collected from pups, except a, samples collected from both pups and dams; b, samples for RNAseq also collected from both pups and dams; c, serum collected at sacrifice from both pups and dams; d, IgAseq also performed in both groups of pups

experiments were selected by the sPLS model (Supplementary Fig. 4a, b). Use of these taxa provided high goodness-of-fit (within sample $r^2 = 0.77$) and statistically significant model coefficients, at $\alpha = 0.05$, and explained 31% of the sIgA variation from the transfer experiment. Three Lachnospiraceae family OTUs were negatively associated with ΔsIgA (Supplementary Fig. 4c), including *Clostridium hathewayi*, previously associated with reduced cytokine production in a murine colitis model[22]. Three OTUs showed PAT-dependent positive associations with ΔsIgA: Rikenellaceae, Enterobacteriaceae, and Lachnospiraceae/ Blautia. The control-dependent ΔsIgA signature identified a *Bifidobacterium* sp. and a Clostridium genus (in the Erysipelotrichaceae family). In total, these analyses identify taxa individually associated with the altered sIgA phenotypes.

**PAT induces conserved microbial network topology alterations**. Next, to define the effect of PAT-induced perturbations on the structure of microbe–microbe associations, we developed a network model based on combining analysis of the dosing experiment (Fig. 1) and transfer experiment (Fig. 4). We inferred networks using SPIEC-EASI and used the stability metric to rank confidence in individual edges for the transfer experiment networks. Using this ranking to generate an ROC curve, we compared the ranked edges from the transfer experiment against the 'true' corresponding edges from the dosing experiment using the set of OTUs common to both experiments. Between the two experiments, we could compare Control-Control, PAT1-all PAT, and PAT3-all PAT relationships. Assessing the maximum harmonic mean of Precision and Recall (the $F_1$ score), more edges were recovered than expected by at-random prediction (dotted red line), under all conditions (Supplementary Fig. 4d). Overall, edges in the control mice were more likely to be recovered in the transfer experiment group ($F_1 = 0.75$). PAT vs PAT1 resulted in higher edge recovery ($F_1 = 0.72$) than PAT vs PAT3 ($F_1 = 0.62$),

the expected outcome of additional antibiotic pulses. These results indicate that together with composition, condition-dependent association networks were largely conserved across the fecal transfer experiments, and lead to the hypothesis that the networks themselves help establish the new communities that drive the observed phenotypes.

To further examine this hypothesis, we considered the global topological properties of the inferred networks. Using graphlet correlation distance as a global measure comparing networks, and using isometric MDS as an analytic tool, we inferred low-dimensional embedding of the microbial association networks (Fig. 5). Notably, as expected, there was linear separation between the two experiments, but we also observed linear separation between the Control and PAT conditions across the experiments. These findings indicate that while transfer of distinct entire microbial communities affects both composition and inferred network topology in the recipients, comparing higher order network structures also can reveal global, transferable patterns of association.

**Roles of keystone taxa in the community dynamics**. We next hypothesized that antibiotic interventions promote network fragility by altering representation of keystone taxa that mediate multiple intra-community fitness interactions. As such, we asked whether overall network robustness mediated via the keystone taxa was affected by the differential exposures. To test this hypothesis, we used natural connectivity as a stability metric of the inferred networks after simulated network "attacks". Considering theoretical keystone indices, we assessed network stability after removing OTUs ordered by degree or betweenness centrality (Supplementary Fig. 5a), or at random (as a control) (Supplementary Fig. 5b). In the random attack setting, overall networks inferred from the dosing (Dose) experiment (Fig. 1) were more robust than transfer (Trans) experiment (Fig. 4)

networks. In the degree and betweenness-based attack schemes, the Dose-PAT3 networks were relatively fragile, and were at the same level as both groups from the transfer experiment (Supplementary Fig. 5a–d). Within the groups from the dosing experiment, Control and PAT were equally robust in the random and degree-based removal approaches, but PAT1 was more fragile to betweenness-attack schemes. These results indicate that if experimental interventions (i.e. antibiotic exposure or transfer to new hosts) targeted bottleneck keystone taxa, communities would be destabilized.

Finally, we also considered the particular OTUs identified as potential keystone taxa in the SPIEC-EASI-inferred association networks. For each network, we identified the three highest-scoring OTUs that were both hubs (having high node degree) and bottlenecks (as characterized by the highest betweenness centrality) in the network (Supplementary Fig. 5e). Despite the differences between the experiments revealed by the network topology and robustness analysis, the taxonomic identities of these species were conserved within treatment type, across both the dosing and transfer experiments; in Controls, these were Lachnospiraceae, *Blautia*, *Blautia producta*, and Enterobacteriaceae, but only S24-7 in PAT1. Taken together, these results allow the prediction that perturbation of these taxa impacts the stability of the overall community, especially since their keystone network attributes were conserved even after transfer into GF mice.

## Discussion

These mouse studies show that even a single macrolide antibiotic course with dosing that models those used in human children has effects well beyond the actual period of antibiotic exposure (see Fig. 6) for schematic of all experimental details). In this murine model, we administer the first course of the macrolide, tylosin, which has a similar mode of action to the human macrolide antibiotics erythromycin, clarithromycin, and azithromycin, to developing pups through their nursing dams. This 5-day course is unlike other studies in which supertherapeutic levels of antibiotics were administered experimentally for several days to weeks to perturb immunological development[23]; we dosed to achieve the gastrointestinal levels that are typically observed in children given a macrolide to treat acute infections. The amount of antibiotic that reached the nursing pups was sufficient to alter microbial richness and structure, demonstrating its antibiotic effect and mimicking effects seen in human children after a single antibiotic course[24]. Using this model, we found that such a brief early-life antibiotic course altered community network topology, greatly perturbing microbial recovery for months. Such exposure, with important bearing on microbiota development, yielded altered ileal gene expression signatures and consequent immunological effects, in an age-specific manner, with much greater effects on pups than on their dams.

The most consistent immunologic abnormalities observed were decreases in the frequency of SI-LP CD4+ IL17A+ lymphocytes and in intestinal sIgA secretion. Although we did not see major differences in other T-helper cell populations, nor specifically in expression of Th2-associated cytokines, early-life microbial perturbations may alter the cellular plasticity of CD4+ populations and may skew T-helper cell polarization, this point needs to be addressed in future studies.

Since Th17 cells may induce sIgA production[25], one possibility is that the Th17 abnormality, upstream of sIgA, might directly reflect the microbiota shifts. Alternatively, both phenotypes might be secondary to a common upstream effector. Segmented filamentous bacteria (SFB) ("Savagella") are microbes

with important immunomodulatory properties[26]; however, the transfer study shows immunological effects that are SFB-independent. Although in this study the immunologic and microbial phenomena were observed in C57BL/6 mice, a recent publication demonstrated similar phenotypes after early-life, macrolide-induced microbial perturbation in a non-obese diabetic murine model[16].

Macrolides may have direct immunomodulating effects[19]. However, their limited effects in germ-free mice, demonstrated by ileal gene expression analysis and T-helper cell profiles (Fig. 3), provide evidence that a microbiota is required for the effects we observe. These observations suggest that the main mechanism for the immunomodulating macrolide actions is via selective alterations in microbial composition. We interpret these findings as providing evidence that the immunomodulation effects in our model are primarily due to antibiotic-induced microbial community perturbations and not to direct tissue interactions, findings that may have relevance to prior studies[19, 27, 28].

This body of work demonstrates how early-life macrolide treatment shapes the ecology of the microbiota. The bloom in *Akkermansia muciniphila* was consistently observed after antibiotic treatment in each experimental modality, this phenomenon may be due to a loss in competitor mucin-degrading bacteria[24]. An *Akkermansia* bloom has been seen in other PAT models[12, 16] and has been associated with divergent host outcomes including improved metabolic phenotypes[29], or alternatively, increased susceptibility to colitis.[30, 31] The family Enterobacteriaceae was consistently increased after antibiotic treatment in this study, similar to previous observations after antibiotic treatment[32] and may be important mediators of downstream immunologic phenotypes. The S24-7 family, with the recent proposed name "*Candidatus* Homeothermaceae"[33], is dominant in control groups across experiments and is substantially decreased with antibiotic treatment. Members of the S24-7 family are targeted by sIgA[34, 35], and their loss may alter sIgA labeling of the microbiota. The prominence of these taxa may reflect ecologic perturbations that allow opportunistic organisms to bloom, and/or they may play causal roles in the observed phenotypes. Since blooms in these taxa have been associated with phenotypic alterations in prior studies, this point needs resolution.

We show the effects of early-life antibiotic treatment on bacterial ecological succession after transfer to a microbe-naïve host. The distinct community structures acquired at transfer that persist for the experiment's duration indicates the importance of founder effects from prior hosts. These in turn reflect the effects of the prior antibiotic-induced perturbations on ecological resilience and stability during succession[36, 37]. Such concepts also may be relevant to the transfer of microbiota at birth from mothers with recent antibiotic treatment, especially macrolides[13].

Although the effects of three antibiotic pulses (PAT3) were greater, even a single brief early-in life course (PAT1) yielded multiple immunological effects, detected > 40 days after the exposure ended. Whether the exposure irrevocably altered immunological development or whether the effects reflect the persistence of an altered microbiota were not yet defined. However, the transfer experiment indicated that an altered microbiota was sufficient to convey delayed sIgA secretion, monitored over the course of the experiment, as well as immunological abnormalities 11 weeks after the conventionalization, in contrast to prior transfer studies with more brief observation periods[38, 39].

Sex-specific differences in taxonomic profiles observed in the PAT-conventionalized group highlight the potential influence of

sex hormones on the microbiota, which has been previously demonstrated in a mouse model of spontaneous type 1 diabetes and several human and other animal models[40–42], and should be further studied. Conventionalization of germ-free mice with either control or PAT-perturbed microbiota shows the rapidity of developing altered phenotypes, with an sIgA difference emerging 12 days later. These studies confirm that unperturbed microbiota are critical for inducing normal sIgA expression[43], and the IgA-SEQ studies provide definition of the relevant taxa.

All antibiotic exposures are not equal; our work shows differential effects based on host age, sex, and number of courses. Nevertheless, the antibiotic-induced phenotypes remained consistent across experiments, which increases the confidence in our findings. We propose that after antibiotic treatment in early life, key taxa are diminished or lost and replaced with taxa that are less interactive with the intestinal epithelium, reducing homeostatic expression of innate microbial sensors and ultimately leading to reduced Th17 differentiation, and secretory and systemic IgA expression.

Out-of-network analysis indicates differences in inferred microbial topology and supports the notion that antibiotic treatment decreases stability. From the perspective of ecological dynamics, this decrement partially explains why the observed decreases in diversity are stable both following antibiotic pulses and following transfer of antibiotic-perturbed microbiota to GF mice. Consistent with prior observations[44], the network analysis suggests a model in which antibiotic effects are transmitted to the host only after perturbation through the network of interactions that comprise and enable stable gut microbial ecology. Further studying the interactions between key taxa identified here can better uncover the complex mechanisms underlying responses to antibiotic exposure[45]. The taxa that may potentially influence phenotype are natural targets for direct intervention, to either complement or mitigate antibiotic effects in future experiments, and to probe the mechanisms behind interactions within the microbial ecosystem. In addition, the relationship between putative keystone species and immunological roles will need to be addressed in a future study.

The observations seen in these studies, including changes in microbial populations, ileal gene expression, and cellular responses, demonstrate the overall impact of one early-life macrolide course on host microbial community networks and the conjunction with host immunity. The translation of these impacts on host physiology and health is not completely defined in this study; however. previous studies demonstrate that perinatal antibiotic exposure leads to increased susceptibility to allergic asthma and hypersensitivity pneumonitis[46, 47], abrogates tolerance to dietary antigens[48], and in non-obese diabetic mice accelerates the development of type 1 diabetes[16]. This work highlights potential downstream effects of early-life antibiotic treatment on microbial community networks and its relationship to host immunity and these principles are potentially translatable. Nevertheless, murine models are not always a proxy for effects in humans[49].

Particular antibiotic classes, including both macrolides and β-lactams, have differential activities against anaerobic bacteria[50]. These studies provide evidence that even a single macrolide antibiotic course early in life leads to long-lasting effects on intestinal microbial community structure, network topology, and immune phenotypes. An antibiotic-perturbed microbiota alters microbial succession, and is both necessary and sufficient for impaired immune phenotypes. The dynamics of the ecological and physiologic changes suggest the importance of network stability, which may be especially at risk in early life.

## Methods

**Mice.** All experiments were conducted with C57BL/6 mice. To generate the SPF litters for the dosing and germ-free experiments 7-week-old males and females were purchased from Jackson Laboratories (Bar Harbor, ME) and bred. The first litter of each dam was randomly assigned to an experimental group. Litters were weaned at 3 weeks of age, and segregated by sex and treatment group, 2–7 litters per treatment group, with a target sample size of 3–10 mice per treatment group based on prior studies. Germ-free mice were bred at NYUMC and maintained germ-free state. For the transfer experiment 5-week-old male and female germ-free C57BL/6 mice were colonized with PAT- or control microbiota conventionalizing them. Mice were fed a standard 10% kcal fat rodent chow (PicoLab Rodent Diet 20; LabDiet, Brentwood MO), and maintained on a 12-h light/dark cycle and allowed ad libitum access to food and water. All mouse experiments were approved by the New York University School of Medicine Institutional Animal Care and Use Committee (IACUC protocol no. 160613) and complied with federal and institutional regulations.

**Antibiotic exposures.** The antibiotic regimen used was based on the prior PAT protocols[12, 16]; briefly, tylosin tartrate (Sigma Aldrich) was dissolved in non-acidified water at a concentration of 333 mg/l to achieve a dose of about 50 mg per kg body weight per day. This dose is therapeutic for rodents[51] and mirrors the concentrations commonly used for human children for an acute infection[52].

For all experiments dams and their litters were randomly assigned into antibiotic or control groups. In the dosing experiment control mice received non-acidified water, PAT1 and PAT3 groups received one tylosin course at day 5 of life for 5 days, through their mothers' milk. The PAT3 cohort received an additional two pulses at days 27 and 36 for 3 days. In the germ-free PAT experiment, mice received autoclaved water alone (control) or containing tylosin at day 5 for 5 days. Autoclaving tylosin did not reduce its antibiotic potential. To assess the antibiotic activity of autoclaved tylosin water, we autoclaved tylosin in LB at three concentrations: 666, 333, and 166.5 μg/ml. Commensal bacteria that were macrolide-sensitive were inoculated to LB plates alone, or containing autoclaved tylosin, or filtered-sterilized tylosin, or LB alone. There was substantial bacterial growth on the LB alone plate, while LB plates with either filter-sterilized or autoclaved tylosin showed no growth on any plate, at all three antibiotic concentrations. We also used the autoclaved water in the exposure of conventionalized, specific pathogen free (SPF) mice to control for the effect of autoclaving tylosin in the germ-free mice. For the transfer experiment, mice were exposed to a single tylosin pulse at day 5 for 5 days, and cecal contents were collected from day 12 mice for transfer into germ free recipients; none of the recipient mice received any antibiotics. The investigators conducting subsequent assays were unaware of the study groups to which individual animals belonged.

**Microbiota transfer.** For the transfer experiment, cecal contents were collected from donor mice that received PAT or non-acidified water (control) between P5-10 and sacrificed at dol 12. The contents were divided, and 1/3 was immediately placed in pre-reduced anaerobic dental transport media (Anaerobe Systems, Morgan Hill CA), as described[38] and frozen at −80 °C. Upon thawing under anaerobic conditions, the cecal contents from 3 mice per group (PAT or control) were pooled and further diluted in dental transport media; 100 ul of the suspension was transferred to 5-week-old C57BL/6 germ-free mice via oral gavage. Paired mice received either the PAT or control suspensions in three cohorts based on availability of germ-free animals in our facility; all recipients were caged in the same animal treatment room to minimize environmental variation.

**Measurement of intestinal, fecal, or serum IgA.** Serum samples were diluted in 1× phosphate-buffered saline (PBS) at 1:10,000. Intestinal or fecal samples were resuspended in 1× PBS at a concentration of 50 mg/ul by weight, and supernatants were stored at −20 °C. Total serum, fecal, and intestinal IgA were measured using the mouse IgA Elisa kit, according to the manufacturer's instructions (Bethyl, Montgomery TX, USA). For the dosing and transfer experiments, IgA data were collected from the repeated fecal sampling. The absorbance was measured at a wavelength of 450 nm using the Dynex MRX TC Revelation microplate reader (Dynex Technologies, VA, USA).

**Statistical approaches.** For the transfer experiment data, a random effects analysis of variance was done to test for the difference between the recipients of PAT and the control inocula[53]. In that analysis, treatment and sex were considered fixed effects and cohorts were considered random. SAS software and R were used to perform the tests and calculate the estimates. LEfSe was used to discover significantly differentially abundant bacteria species between two or more groups, using sample relative abundances. DESeq2 was used to normalize and compare differences in host ileal transcriptomics between treatment groups.

**IgA-Seq.** Using specimens from the transfer experiment, we performed IgA-SEQ, essentially as described in Palm et al.[34]. In brief, we sampled from three time points, day 29 (fecal), 76 (fecal), and 77 (cecal), with 12, 11, and 8 samples tested, respectively. From specimens that had received the PAT or Control inoculum, we saved three fractions: a pre-fractionation sample (pre), and fractions that were

either IgA$^+$ or IgA$^-$ by cell sorting with an anti-IgA antibody, following the procedure of Palm et al. After DNA extraction from the fractions, library preparation and 16S rRNA sequencing using V4 primers, as described[38] using the Illumina MiSeq platform, nucleotides were quality filtered with a Phred score < 20 and chimeras were removed. Filtered reads were clustered into 97% identity OTUs using UCLUST program, followed by taxonomy assignment using the RDP Classifier. The OTU absolute abundance table was extracted from the pipeline for further analysis including generating mean taxonomic abundance plots, calculating the IgA coding index (ICI)[34] of each taxa as follows (relative abundance IgA$^+$ fraction/relative abundance IgA$^-$ fraction), and performing statistical tests using the LEfSe tool[54]. We recognize the limitations of our methods for detecting significantly differently abundant taxa and understand the discord between reliable normalization and testing approaches[55, 56], which is why we sought to use more established statistical methods.

**Isolation of splenic and intestinal lamina propria lymphocytes.** Briefly, the small intestine was placed in Hank's Balanced Salt Solution (HBSS) buffer supplemented with 10% fetal bovine serum. Peyers patches were excised and tissue was placed in digestion media containing 1 mM DTT, 1 mM EDTA in calcium/magnesium-free HBSS supplemented with 2% fetal calf serum and subsequently treated with Collagenase IV/Dnase digestion mix (0.5 mg/ml of collagenase IV and 200 μg/ml of Dnase). Lymphocytes were enriched using a 44/67% discontinuous Percoll (GE Lifesciences, Pittsburgh PA) gradient. Splenic lymphocytes were treated with ammonium chloride and washed. Cells were stimulated with phorbol 12-myristate 13-acetate and Ionomycin for 4 h at 37 °C in the presence of brefeldin A (Golgi-Plug; BD Bioscience, San Jose, CA). Following stimulation, cells were stained with LIVE/DEAD Fixable Aqua (ThermoFisher Scientific, Waltham, MA), and the following antibody/fluorophore combinations APCC7-CD45, TCRβ-FITC, CD4-V500 (BD Bioscience), CD8-BV650, Foxp3-PECy7, IL17-PE, IFNγ-FITC (eBioscience, San Diego, CA), and fixed with fix/perm (eBioscience), were used according to the manufacturer's instructions. Cells were acquired on an LSRII flow cytometer (BD Bioscience) and analyzed with FlowJo software (Tree Star, Ashland, OR); 100,000 events or greater were collected for each sample. Samples with yields less than 10,000 viable events were excluded from analysis.

**RNA extraction and nanostring analysis.** RNA was extracted from ileal tissues using the Qiagen miRNAeasy kit (Qiagen, Valencia, CA), samples were measured using the nCounter GX mouse immunology (NanoString Technologies, Seattle WA). Counts were normalized using DESeq2 and log$_2$ transformed. Significance between groups was measured using the Anova (aov) function from the stats R-package, and post-hoc analysis performed using TukeyHSD. P-values were corrected for multiple comparisons, based on the False discovery rate (FDR), with significance considered by q-value < 0.05. Heat maps were generated using the pheatmap package. All analysis was performed using custom R functions.

**RNAseq.** RNA was extracted as above, and sample libraries were prepared using poly (A) enrichment and the TruSeq v2 stranded mRNA kit (Illumina, San Diego, CA). 1 × 50 bp single-end libraries were generated using the Illumina HiSeq 2500. The 16 ileal samples yielded a total of 464,187,920 reads with a mean count of 28,970,166 + 5,187,775 (mean + SD) per sample. Samples were processed as follows: bases with a Phred score < 20 and reads < 20 bp were removed using cutadapt v1.9.1. Surviving sequences were then passed to SortMeRNA v2.1 to remove eukaryotic, archaic, and bacterial ribosomal RNA (rRNA) sequences. Reads were aligned to the mouse GENCODE GRCm38 (M12 release) genome using STAR v2.5.1b[57], and the read summarization program featureCounts was used to count mapped reads against annotated genes. Differential expression analysis was performed with the R-packages DESeq2[58]. Differentially expressed pathways and functions were interpreted using Ingenuity Pathway Analysis (IPA, QIAGEN Redwood City, http://www.ingenuity.com).

**DNA extraction and amplicon library preparation.** DNA was extracted from fecal, ileal, and cecal samples using the Mobio 96-well extraction kit following the manufacturer's instructions. For amplicon library generation, the V4 region of the 16S rRNA gene was amplified with gene-specific primers, as described[38]. The reverse amplification primers contained a 12-base pair Golay barcode to support pooling up to 864 samples. Amplicons were prepared in triplicate, pooled, and quantified. The 254 bp V4 region was sequenced using the Ilumina MiSeq 2 × 150 bp platform. OTUs were picked using GreenGenes 13_8 for reference, using the open reference picking strategy.

**Microbial community analysis.** Briefly, the amplicon read processing pipeline, QIIME 1.91, was used[59]. Nucleotides were quality filtered with a Phred score < 20, reads were clustered into 97% identity OTUs using UCLUST program, followed by taxonomy assignment using the RDP Classifier and chimeras were removed using ChimeraSlayer.

Samples with a sequencing depth < 1000 were excluded and OTUs with a relative abundance < 0.01% were removed. The phylogenetic tree and abundance tables generated were used to calculate unweighted and weighted UniFrac β-diversity indices[60]. The filtered OTU abundance table was used for

further analysis using the phyloseq package in the R statistical programming environment[61]. Inter-group UniFrac distances were averaged over independently drawn sample pairs (subsampled without replacement and replicated 999 times).

**Total 16S and SFB quantitation.** Total 16S rRNA gene copies were quantified by qPCR in fecal samples using the forward primer 785F(5′-GGM TTA GAT ACC CBG GTA GTC C-3′) and the reverse primer 907R(5′-CCG TCA ATT CMT TTG AGT TT-3′), degenerate primers spanning the V5 region of the 16s rRNA gene. SFB were quantified using the 736F/844R primer pair[62]. 16S and SFB standards were included in each run to obtain absolute quantitation. qPCR was performed with the LightCycler 480 SYBR Green I Master mix (Roche) and run in a LightCycler 480 system (Roche).

**Inference of microbial association networks.** For each treatment group within the dosing (Fig. 1) and transfer (Fig. 4) experiment sets, we learned microbial association networks using the Sparse InversE Covariance estimation for Ecological ASsociation Inference (SPIEC-EASI) framework[63]. Nodes in each network correspond to OTUs and edges correspond to direct signed interactions between OTUs given each environment. We ran SPIEC-EASI in neighborhood selection mode and performed model selection via StARS[64], using a variability threshold of 0.05% to increase precision[63].

**Analysis of microbial association networks.** We followed a prior procedure[65] to assess the overall similarity of the five different association networks. We enumerated all induced subgraphs (graphlets) composed of up to four nodes in each network and for each node, recorded the frequency of participation in each subgraph. We used the Spearman correlation among 11 non-redundant subgraph frequencies (orbits) across all nodes as robust and size-independent network summary statistics, as described[66]. Pairwise distances between entire networks were computed by using the Frobenius norm between the correlation matrices (graphlet correlation distance)[66]. To achieve a low-dimensional description of network similarities, we embedded these distances in Euclidean space using classical MDS. We also assessed the robustness of the different microbial association networks to random and targeted node removals ("attacks")[67] using natural connectivity[68], as a graph-theoretic measure of global network connectivity that reliably measures network robustness. We measured how natural connectivity of the microbial network changed when nodes are sequentially removed from the network. We applied three different types of network attacks: (1) random node removal (using $N = 30$ random orderings), (2) node removal ordered by betweenness centrality, and (3) node removal ordered by node degree (number of neighbors). Betweenness centrality[68] measures a node's centrality in a network by calculating the number of shortest paths from all nodes to all others that pass through that particular node.

**Sparse and compositionally robust multilevel PLS regression.** We used our previously published compPLS framework[65, 69] to detect associations between specific taxa in fecal microbiota communities and longitudinally measured host phenotypes. Briefly, to overcome the detection of statistically spurious associations, we (1) center log-ratio (clr)-transformed the OTU relative abundance data; (2) applied a variance decomposition to extract within-subject variation; and (3) estimated a sparse linear model via sparse Partial Least Squares (sPLS) regression for detecting associations between a sparse set of multi-collinear features (OTUs) and responses (host covariates)[70]. We selected the OTUs that were common to both the dosing and transfer experiments, which yielded comparable perturbations, and further restricted the data set to those taxa present in > 20% of the samples in both experiments. Finally, we restricted our sample set to include only those with corresponding measurements of fecal IgA. This resulted in a set of 53 taxa at $n = 96$ and $n = 57$ samples for the dosing experiment (Fig. 1) and the transfer experiment (Fig. 4), respectively. We applied multilevel-sPLS to the dosing experiment data set, using stability selection to obtain confidence scores on model coefficients at a fixed sparsity level[65]. We used these scores to iteratively filter OTUs from the model, selecting the set that maximized the $r^2$ of the out-of-sample prediction of IgA measurements from the transfer experiment between 6 and 43 days post-transfer. We computed biplots from the sPLS model, with sample scores colored by (scaled and centered) changes in IgA responses (ΔIgA) and associated OTUs, represented by a loading vector colored by taxa family.

**Data availability.** The RNAseq data that support the findings of this study have been deposited in ArrayExpress database (www.ebi.ac.uk/arrayexpress) with the accession code E-MTAB-5101 (http://www.ebi.ac.uk/arrayexpress/experiments/E-MTAB-5101/), and the 16S rRNA metagenomic data has been deposited in QIITA (https://qiita.ucsd.edu/) with the identifier 10527 (https://qiita.ucsd.edu/study/description/10527). Ileal Nanostring data have been deposited in NCBI's Gene Expression Omnibus and are accessible through GEO SuperSeries accession number, GSE98292 (https://www.ncbi.nlm.nih.gov/geo/query/acc.cgi?acc=GSE98292).

We generated reproducible Jupyter and RMarkdown notebooks that contain the commands and data used to generate the main figures found within the manuscript (https://github.com/blaser-lab/Paper-Ruiz-2017). The authors declare that all other

relevant data supporting the findings of this study are available in the article and its Supplementary Information files, or from the corresponding author upon request.

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

## Acknowledgements

We thank the NYULMC Human Microbiome Program laboratory, NYULMC Genome Technology Center, Peter Meyn, Paul Zappile, Susan Joseph, the Colton Center, and the flow cytometry core for assistance. These studies were supported by DK090989 (NIH), the Diane Belfer Program in Human Microbial Ecology, Knapp Family, Ziff family, and C&D foundations. Sequencing performed at the NYUMC Genome Technology Center (P30CA016087).

## Author contributions

V.E.R and M.J.B. designed experiments and interpreted the data. A.O., I.E., X.Z., V.E.R., T.B., B.J.M. and T.I. performed the experiments. V.E.R. and T.B analyzed gene expression and microbiome data. L.B. Z.D.K, R.B. and C.L.M. contributed to the cor-relation, network, and statistical analyses. K.C. and G.P.P. provided critical insights. V.E. R. and M.J.B. were responsible for writing this manuscript, as reviewed by all authors.

## Additional information

**Competing interests:** The authors declare no competing financial interests.

