## [Peer Review File · Nature Communications]

Reviewers' comments:

Reviewer #1 (Remarks to the Author):

The manuscript by Ruiz et al compares the effect of a single dose of the antibiotic tylosin to that of multiple administrations on the development of mice microbiota over time. The authors conclude that the single dose is as disturbing as multiple doses but only in young mice as indicated by effects on microbiota composition, IgA coated microbes and ileal gene expression. Moreover, the authors showed the impact of the disturbed microbiota by its transfer to germ-free mice. Finally, network calculations were performed to further expand on the disturbances in the microbiota.

This is an important study that extends earlier work of the authors and others. However, there are three important issues to address next to some other points

Major issues

1. There are large differences in the effects of broad spectrum antibiotics on the human intestinal microbiota. Here one antibiotic is used, the macrolide tylosin. Moreover, the selection of this antibiotic is not sufficiently rationalized (targets effect in children?). The authors should address this point by either analyzing other antibiotics or clearly describing the limitation of this study.
2. All experiments were performed with C75BL/6 mice – it has been well documented that different mouse lines have different microbiota – partly affected by the supplier (see Xiao et al Nat Biotech 2005). Hence, it would be important to replicate the study in other lines than C75BL/6 mice.
3. In previous studies these and other authors showed the impact of antibiotic administration on weight and other physiological properties. Moreover, strong correlations have been reported in human studies. Remarkably, this is not addressed here. Moreover, the study suggests that the single use of tylosin has immunological effects but these are not translated into a phenotype. It is essential to show a health impact of the tylosin use.

Other points

1. The authors do not indicate sufficiently that this is a mouse study and they use only one antibiotic, tylosin. This should be clearly indicated in the title, abstract and results.
2. Fig 4D shows the transfer of tylosin-treated microbiota in GF mice but the effects in males and females are quite different – this needs to be explained.
3. The GF experiment with tylosin used autoclaved tylosin (line 448 etc) – the authors indicate that this does not affect its potency as results not shown – these data should be included and performed with a really sensitive antibiotic test. In addition, the authors should explain why they did not simply filter sterilized the antibiotic.
4. Line 428: the authors describe a 'target sample size of 3-10 mice per treatment group' – so how many were there really per group? This should at par with studies where at least 5-6 mice are used in each group – if not this should be clearly mentioned in the legends of the Figures.
5. The PIEC-EASI approach is a relative new computational method and it is good to see this applied here. However, it receives a lot of attention at the costs of the biology – for instance the effect on gene expression is not well discussed. This should be corrected. Moreover, I also suggest to tune down the entire description of key stone species in absence of biological data supporting their function.
6. The term ecological volatility is used (line 374) and I wonder what is really meant with it. Similarly, the term 'triangulate' is used (line 389) but seems a bit out of context here.

Reviewer #2 (Remarks to the Author):

Summary

Early childhood exposure to antibiotics is associated with increased risk of immune diseases such as asthma, allergies and inflammatory bowel diseases, and a hypothesis is that these pathologies are the results of antibiotic-altered intestinal microbiota [Blaser & Falkow, *Nature Reviews Microbiology* (2009)].

While previous studies showed that massive antibiotic exposure can perturb the immunological development in mice, the authors here study the role of a single pulse antibiotic treatment (PAT) course on the intestinal microbiota and on the host developing immune system.

The presented results show that: (1) a single PAT course in early childhood is enough to produce durable alterations of intestinal microbiota and of immune system (T-cell populations and secretory IgA expressions), (2) a PAT perturbed intestinal microbiota is sufficient to transfer delayed secretory IgA expressions, and (3) early childhood exposure to antibiotics has lasting and transferable effect on microbial community network topology.

Pros.

Overall, the study is well explained and well written. Also, I don't see major methodological flaws in the experiments.

In most part, the obtained results confirm already hypothesized / measured effects of early childhood exposure to antibiotics, with the key result that even a single PAT course can induce long lasting impairments of host immunity.

I also like the analysis of microbial association networks, although there are issues with how randomized algorithms are used (see point 4 below).

Cons.

1. The study is based on a small number of mice (the smallest group having only 3 mice). While this must be balanced with the cost of the experiments (gene expression, capturing meta-genomes), this still reduces the confidence in the reported results, which is problematic given the potential importance of the presented results.
2. The paper should position itself with respect to other recent papers about antibiotic (early) use and its consequences on microbiome [e.g., Gustafsson et al., *Journal of Immunology* (2016)]; are the results consistent? While the study mentions the differences between control and antibiotic-altered intestinal microbiota in terms of increased/decreased abundance of specific microbe species, it somehow fails at explaining if the changes are expected or not (e.g., is the species known to be associated with a diseases, or maybe its abundance also changed in other similar studies). There is only one small paragraph with one citation (page 14) which quickly discuss such relevance, which should be extended, in particular for the keystone taxa, which the authors suggest to be necessary and sufficient to explain the immunological changes.
3. The authors should also better explain how their results could be translated to the human case; the usage of mouse for the study of human gut microbiota has already been questioned (e.g., Nguyen et al., *Disease Models and Mechanisms*, 2015), and changes that last more than 40 days may be seen as long term changes for mice, but not necessarily for human.
4. In the analysis of microbial association networks, the authors use multidimensional scaling (MDS) to embed networks obtained from different conditions in 2D space according to their pairwise distances (computed using GCD), and to observe that in the 2D space, there are linear separations between networks from different conditions. Given that MDS is usually a randomized process, this experiment should be repeated at least 10 times and the results should be average over different runs to account for randomness. Similarly, the authors measured the susceptibility of these networks to random node deletions, without mentioning the percentage of deleted nodes, or the numbers of independent runs that they performed.

Reviewer #3 (Remarks to the Author):

This manuscript describes the effects of a single and moderate course of antibiotics on the gut microbiota and immune development in mouse. It is shown that the effects of antibiotics treatment in early life leads to changes in microbiota and immunity that are qualitatively somewhat different and more long-lasting than the effect of a similar antibiotics treatment in adults. Comparisons to germ-free control mice and fecal transplants are used to demonstrate that the effect of antibiotics on immune development are persistent, transferable, and mediated by the microbiome. Here the paper proposes a mechanism where the antibiotics affect the species ecological network topology, making it more fragile and less resilient. The experimental design, data, statistical analysis and argumentation are robust. The manuscript is written in a very clear and concise manner; the results section is clear yet somewhat technical considering general readership.

* Major claims

This paper makes the following major claims (in a mouse model):

1) A single pulsed antibiotic course in early life has significant impact on host immunity.

This was shown to lead to a decrease in T-cell populations, intestinal lymphocytes and secretory IgA expression serum/fecal/intestinal, and expression of genes related to immunity (the effects were also greater with 3 repeated antibiotics courses compared to a single course). Changes in immune response were not observed in germ-free controls, indicating a critical role of the microbiome in mediating these effects. Here it should be noted that the observed immunity variables were readily low in GF mice and were not further affected by antibiotics. Perturbations in the microbial community were both necessary and sufficient for observed immune response, which was further established by showing that the immune development was delayed in a fecal transplant experiment in germ-free mice who received microbiota from the antibiotics treated mice.

2) Antibiotics perturbation led to particular changes in the topology of the species co-occurrence network.

This is shown by demonstrating that the species network in the antibiotics treated mice is more fragile; fragility is indicated by quantifying changes in network centrality and betweenness following species removal. The effects of species removal are more remarkable in the antibiotics treated mice and the network topology changes in a way that indicates increasing fragility.

3) The antibiotics induced changes in early life microbiome and host immune phenotypes are long-lasting, persistent, and transferable to a new host.

It is shown that the transferred microbiota was as such sufficient to convey a delayed immune responses up to 11 weeks after the transfer to a new host. The gene expression changes and changes in the immune response were age-dependent and more remarkable in the pups compared to adult mice, and the community recovery was faster in adult mice than in the pups.

* Novelty and interest

- Earlier mouse studies focusing on early life immunity development have investigated the effect large antibiotics doses. The novelty in the present study is that it investigates the effects of a smaller moderate course, which is said to mimic early life antibiotics in humans and hence expected to provide useful hypotheses of the role of antibiotics in human health. This is of wide interest as similar well-controlled human studies are more challenging or impossible to conduct for obvious experimental and ethical reasons. The study design and results provide new experimental evidence that supports the earlier (and appropriately cited) hypothesis that microbiome mediates the effects of antibiotics on host immunity.

- The idea that antibiotics impact species networks is not new as such (one relevant reference that could be added is PMID:20440275). However, the targeted analysis that provide details on these effects following a moderate early life antibiotics course is new, albeit limited in scope as it focuses on taxonomic co-occurrence networks rather than the actual molecular networks and temporal dynamics.

- New experimental evidence is also presented for the claim that the changes in microbiome are long-lasting, persistent and transferable, even after a moderate course of antibiotics, and that changes in the microbiome are critical for the immune development. The role of Akkermansia in iGA expression and immunity is specifically discussed in this context. The recent review by Derrien et al on the Akkermansia and host immunity (PMID:26875998) is relevant here and could be cited.

- Overall, appropriate references are provided throughout the paper.

* Quality of evidence

- The observations are systematic and robust, and different parts of the analysis are very well complementary. The analysis has central importance for a better understanding of the long-term health implications of early life antibiotics use (effects on host immunity), and in particular its microbiome mediated nature (established by comparisons to germ-free mice and fecal transplants), including an analysis of the timing (early life vs. adults) and repetitiveness (1 vs 3 courses) of the antibiotics treatment. As such, the study helps to establish the experimental basis for studying how antibiotics impact microbial ecological networks, gene expression and immune response with potentially long-term functional consequences. As expected, the antibiotics effects also tend to be stronger in the group that received multiple treatments. Overall, the study design and experiments provide robust complementary evidence to support the main conclusions.

- The authors suggest that the antibiotics might affect microbiome structure by affecting taxonomic network topology via highly connected keystone species. Interestingly, the identity of the identified keystone species seems to be robust to antibiotics dosing and microbiome transfer. Moreover, the analysis indicates that the antibiotics perturbed networks are more fragile (more sensitive to species removal), and that the network structure is an essential element in the transferable nature of the persisting effects of antibiotics. The network analysis is limited, however, as it is primarily based on taxonomic co-occurrence networks and lacking analysis of the network dynamics within individuals over time. Measures of network topology are only indirect measures of fragility. Therefore, the fragility could be shown experimentally but this would require study designs that are out of scope for this work. This limitation could be stated in the discussion, however. Alternatively, the indicated fragility measures could be a side product of switching community types (PMID:26866806). This could be cited and discussed. As such, the experimental evidence on microbiome fragility is preliminary and does not cover molecular functional mechanisms or temporal dynamics, although the former is being discussed. Gathering further

experimental evidence on how antibiotics treatment affects the network dynamics and molecular function would be interesting but does not seem feasible for the scope of this paper but the limitations could be stated in the discussion.

- The follow-up time are 40-77 days. To me this seems an intermediate rather than long-term follow-up. Some clarification might help.

- Recently, it was proposed that exposure to diverse microbiomes in the living environment is essential for restoring microbial balance after microbiota is depleted. I think there was a very recent journal paper which I can't find now. But there is at least a recent abstract on this topic <https://academic.oup.com/ecco-jcc/article/2961708/P768> - since delayed recovery of the microbiome is a key component in this paper, could the delay possibly be due to the fact that the living environment does not provide the normal microbiota (does not accommodate normal mice, for instance). Ideally, the study design would control also for this factor but it may not be feasible in the scope of the current study. This factor could have a remarkable effect on the results and should be discussed in the manuscript. Was this controlled?

* Impact on thinking in the field

- This work provides a robust and useful benchmark for further studies aiming to elucidate the mechanistic link between early life antibiotics, microbiome perturbation, and later life health issues. It covers relevant aspects of the single antibiotics course in early life, including the effect on microbiome composition and co-occurrence network topology, functional implications related to host immune development, and the demonstration that these effects are age-dependent, persistent over time and transferable to a new host.

- The study design and results provide essential guidance for further and more detailed analysis on the underlying mechanisms. The main contribution of the paper is to gather solid experimental evidence for topical hypotheses that are central for contemporary host-microbiome research, rather than in the development of new methods or concepts.

- The main readers will be researchers who study host-associated microbiomes, including the human microbiome, and in particular those focusing on the microbiome early-life development and function, which is a currently very active subfield within microbiome research. Also other fields of microbial ecology where microbiome manipulation by antibiotics or by other means is an active research topic may benefit from the findings. The results section is clear but rather technical; a schematic figure of the experimental setup and observations would be useful; the study design has various aspects with varying time scales and experimental groups. While reading the manuscript, it was difficult to keep track on all these different aspects, and a figure summarizing them all in a single schematic figure (aimed at a general readership) would be helpful.

* Statistical analyses

- Overall, the statistical analyses have been performed rigorously, including appropriate controls, replication, and sample size. The conclusions are supported by the data. I have some comments:

- The inter- and intra-group divergence is defined as the mean pairwise unweighted UniFrac divergences (as described in Fig. 1 caption for instance). Although this has been sometimes used in literature to quantify group divergence, it is critical to note that the mean of all pairwise divergences within a group is sensitive to the group size, and mean is also a biased measure of the group divergence since the various pairwise dissimilarities are not independent (A vs B; B vs C; A

vs C) and these problems grow rapidly with the group size. One solution would be to select a smaller subset of independent random pairs from the group, or comparing the divergences from the group mean within each group between the two groups. At the minimum, the group sizes in the comparison should be equal to avoid bias coming from the group size. Can you comment if these considerations were taken into account when comparing inter- and intra group divergences?

- DESeq2 and ANOVA are used to identify significant differences between groups in RNA profiling data. This seems suboptimal as recent studies (PMID:26028277 and PMID:24699258) have demonstrated high false discovery rates for the t-test (this will generalize to anova), and for DESeq2 (PMID:28253908) in this context compared to other methods that take better into account the nature of sequencing-based microbiome profiling data. The differences in the performance of the alternative methods were so notable that this issue should be addressed. At the minimum, the possible effects of this choice on the analysis results should be discussed and, ideally, the results would be updated or at least confirmed with an alternative method that has been demonstrated to work well in the recent comparative reviews. Many of those improved algorithms are readily available in R/Python and the calculation are straightforward, so this should be feasible.

* Reproducibility; data and code availability

- Methods description: Sufficient methodological details for reproducing the experiments are provided.

- Data availability: data has been made only partially available; the RNAseq and the 16S rRNA metagenomic data are being made available via standard public repositories. However, I could not check these data sets as they are not open at the time of this review. Therefore, I can't confirm whether sufficient sample metadata is available to repeat the analyses. Data on the iGA profiling seem to be missing as well, unless it is included in the cited ArrayExpress/QIITA repositories that I can't access.

- Code availability: references to custom scripts and open source tools are made but no code has been provided. As the source code of the statistical analyses is missing, many potentially relevant technical details will remain hidden. I would strongly suggest providing the full analysis source code available as supplementary methods to accompany the paper online. The code should be sufficiently well documented so that the manuscript results can be derived from the source data (which is also provided). Ideally, the code could be used to directly reproduce the reported results based on the released data. This effort should be moderate and feasible to implement.

- Availability of other materials: I can't comment on the availability of reagents or other wetlab materials as that is out of my speciality.

Reviewer #4 (Remarks to the Author):

In this manuscript, Ruiz and colleagues address the question of how early-life antibiotic exposure affects the long-term composition of the microbiome and the impact it has on intestinal immune homeostasis. Given the continually emerging data that demonstrates a role for the microbiome in shaping immune responsiveness, and the continuing need for antibiotic courses to treat childhood infections, this reductionist study performed in mouse models provides a useful framework for gaining mechanistic understanding of how these interventions could shape immunity. Despite the

excitement associated with these studies, there are a few concerns about the study design and the overall novelty of these findings.

1. In the Introduction, the authors cite a few papers that have used long-term exposure to broad-spectrum cocktails of antibiotics to interrogate the role of the resident microbiota in immune development and function. The authors rightly state that these doses are not representative of how antibiotics are administered to people, especially children. However, they overlook publications that have addressed how early-life exposure to single antibiotics can influence the outcome of disease in models of allergic asthma (SL Russell & BB Finlay, PMID: 25145536 and PMID: 23333861). The antibiotic dosing used in these manuscripts is different to what is being investigated here, but should be used to contextualize the findings reported here.

2. The antibiotic treatment course used here is very similar to what was reported in a previous publication from the lab of the corresponding author. This increases confidence that the results are consistent (decreased Bacterioides, increased Enterobacteriaceae), but there are some notable differences as well (increased Akkermansia). Akkermansia has mucus-degrading function, and germ-free mice or mice treated with the antibiotic cocktail, have a diminished mucus barrier. It is curious that Akkermansia is increased in this setting – have the authors determined whether the mucus lining is affected by their antibiotic treatment protocol?

3. In Figure 2, the authors characterize immune changes associated with antibiotic treatment. They report that mice that were exposed to PAT have no change in total CD4+ T cell numbers, or the frequency of Treg and IFN γ + cells, but a decrease in the frequency of Th17 cells. It stands to reason that there could be a relative increase in the frequency of another CD4+ Th lineage – have the authors investigated whether there is a skew toward Th2 polarization in the PAT-treated mice? This would be relevant to the observation that germ-free or antibiotic-cocktail treated mice are more Th2-like and could be informative for our understanding of how early life exposure to a diverse microbiota shapes immune polarization.

4. In Figure 2, the authors determine that fecal levels of IgA are diminished by PAT1 or PAT3. Given the recent publication showing microbe-dependent degradation of IgA (Moon, Baldrige & Virgin PMID: PMC4425643) it is important that the data be presented in a way that accounts for this. Are the dams that are shown in the right-hand panels the same dams that gave rise to the pups in the left-hand panels? To provide an extra level of confidence that PAT results in decreased sIgA in pups, the authors could monitor sIgA in pups born to an untreated dam, then monitor sIgA in a second litter of pups that come from that same dam + PAT.

5. It is unclear why the authors chose to focus their analysis on ileal tissue given that the majority of the bacterial biomass is in the colon. These assays could be reported on to demonstrate whether there are tissue-specific differences or if there is global disruption of intestinal immune homeostasis.

6. On page 7, the authors state that “importantly, the profiles for PAT-exposed SPF mice clustered with the GF mice, and not with the SPF controls (Figure 3g), even 40 days after the antibiotic exposure, indicating a persistent effect.” This statement should be discussed and compared to their previous PAT paper where they concluded, “Although the PAT groups largely converged with control over time, differences never fully resolved.” (Ref 13 in submitted manuscript)

7. For the PAT-treatment of germ-free mice, it would be worth providing the rationale that macrolides have been proposed to immunomodulatory in the Results section, rather than waiting to introduce the concept in the Discussion.

8. Following immune characterization, the authors perform comprehensive analysis of the microbiota, ultimately concluding “Taken together, these results allow the prediction that perturbation of these keystone taxa profoundly impacts the stability of the overall community;

especially since their keystone roles were conserved even after transfer into GF mice.” While the amount of computational analysis done to get to this place is appreciated, it begs functional analysis. Can the authors test this provocative hypothesis and relate it to the immunophenotypes characterized earlier?

Reviewer #1 (Remarks to the Author):

The manuscript by Ruiz et al compares the effect of a single dose of the antibiotic tylosin to that of multiple administrations on the development of mice microbiota over time. The authors conclude that the single dose is as disturbing as multiple doses but only in young mice as indicated by effects on microbiota composition, IgA coated microbes and ileal gene expression. Moreover, the authors showed the impact of the disturbed microbiota by its transfer to germ-free mice. Finally, network calculations were performed to further expand on the disturbances in the microbiota.

This is an important study that extends earlier work of the authors and others. However, there are three important issues to address next to some other points

Reviewer #1 Major issues:

Comment/Question 1. There are large differences in the effects of broad spectrum antibiotics on the human intestinal microbiota. Here one antibiotic is used, the macrolide tylosin. Moreover, the selection of this antibiotic is not sufficiently rationalized (largest effect in children?). The authors should address this point by either analyzing other antibiotics or clearly describing the limitation of this study.

Response R1-1: Thank you for this comment. We focused on a macrolide since macrolides are frequently used in childhood as an alternative to β -lactams in clinical medicine. Recent studies have also shown how macrolides have more prominent effects on the microbiota compared to β -lactams. Tylosin was selected because it is a macrolide, mimicking macrolides used in children (Letter reference 1). Our prior work also showed that tylosin had a much more profound effect on the microbiota than did amoxicillin (Nobel et al. *Nature Comm* 2015). All of these points now are incorporated into the revised manuscript (page 3 paragraph 1).

Comment/Question 2. All experiments were performed with C75BL/6 mice – it has been well documented that different mouse lines have different microbiota – partly affected by the supplier (see Xiao et al *Nat Biotech* 2005). Hence, it would be important to replicate the study in other lines than C75BL/6 mice.

Response R1-2: Thank you for this comment. We have previously reported the effects of tylosin in the non-obese diabetic (NOD) mouse model (Livanos et al. *Nature Microbiol*, 2016), which has similar effects on the microbiome and on Th17 cells. We now indicate this point in the text (page 15 paragraph 3- page 16).

Comment/Question 3. In previous studies these and other authors showed the impact of antibiotic administration on weight and other physiological properties. Moreover, strong correlations have been reported in human studies. Remarkably, this is not addressed here. Moreover, the study suggests that the single use of tylosin has immunological effects but these are not translated into a phenotype. It is essential to show a health impact of the tylosin use.

Response R1-3: Thank you for this comment. When we embarked on these studies, the questions we sought to address were specific to immunity and immunologic phenotypes. Our lab continues to address the effect of antibiotic treatment on metabolic outcomes including weight and alterations in body composition, but this was beyond the scope of this paper. Moreover, we are currently addressing the health impact of tylosin-induced microbial perturbations. We are working on two additional manuscripts in which we demonstrate that early-life tylosin treatment increases susceptibility to *C. rodentium* infection and exacerbates Dextran sodium salt (DSS)-induced colitis. Our manuscript already is quite extensive, and the DSS and *C. rodentium* experiments each represent a complete study; we now mention both studies in the Discussion section to indicate that we are studying the health impact from the early life PAT exposure (page 19 paragraph 1).

Reviewer #1 Other points:

1. The authors do not indicate sufficiently that this is a mouse study and they use only one antibiotic, tylosin. This should be clearly indicated in the title, abstract and results.

Response R1-4: Thank you for this comment. We have now modified the title, abstract and results, as suggested.

2. Fig 4D shows the transfer of tylosin-treated microbiota in GF mice but the effects in males and females are quite different – this needs to be explained.

Response R1-5: Thank you for this comment. We now have mentioned the differences in males and females and the role of sex on the intestinal microbiota (page 17 paragraph 3).

3. The GF experiment with tylosin used autoclaved tylosin (line 448 etc) – the authors indicate that this does not affect its potency as results not shown – these data should be included and performed with a really sensitive antibiotic test. In addition, the authors should explain why they did not simply filter sterilized the antibiotic.

Response R1-6: Thank you for this comment. In our germ-free facility, all chow and water is required to be autoclaved before it is introduced into the isolator. We autoclaved Tylosin in LB at three concentrations: 666, 333, and 166.5ug/ml (the drinking water contained 333 ug/ml of tylosin) to assess the antibiotic activity. Commensal bacteria from young mice were inoculated to LB plates alone, or to LB plates containing autoclaved Tylosin, or filtered-sterilized Tylosin. There was substantial bacterial growth on the LB alone plate, while LB plates with either filter-sterilized or autoclaved tylosin showed no growth on any plate, at all three antibiotic concentrations. This experiment documented the potency of the macrolide independent of sterilization technique. We also used the autoclaved water in the exposure of the conventionalized, specific pathogen free (SPF) mice, to control for the effect of autoclaving tylosin in the germ-free mice. The SPF mice treated with autoclaved tylosin showed significant alterations in microbial richness and community structure (Figure 3), and specific immunologic phenotypes that have been shown previously in mice that were exposed to non-autoclaved tylosin. These points now are more fully explained in the Methods section (page 20 paragraph 2-page 21, paragraph 1).

4. Line 428: the authors describe a ‘target sample size of 3-10 mice per treatment group’ – so how many were there really per group? This should at par with studies where at least 5-6 mice are used in each group – if not this should be clearly mentioned in the legends of the Figures.

Response R1-7: Thank you for this comment. We have now modified the figure legends (Figures 1-4 and Supplemental Figures 1-3) to clearly mention the sample size for every group.

5. The SPIEC–EASI approach is a relative new computational method and it is good to see this applied here. However, it receives a lot of attention at the costs of the biology – for instance the effect on gene expression is not well discussed. This should be corrected. Moreover, I also suggest to tune down the entire description of key stone species in absence of biological data supporting their function.

Response R1-8: Thank you for the comment. We discuss the effects of PAT on immunological markers and on gene expression throughout the paper (pages 5-6, page 7 paragraph 1, and page 14 paragraph 3- Page 15 paragraph 1), and have reduced the text about putative keystone species (page 13, paragraph 3 and page 17, paragraph 1).

6. The term ecological volatility is used (line 374) and I wonder what is really meant with it. Similarly, the term ‘triangulate’ is used (line 389) but seems a bit out of context here.

Response R1-9: We now have revised the text, eliminating the terms ‘volatility’ (page 16 paragraph 2), and ‘triangulate’ (page 17 paragraph 1).

Reviewer #2 (Remarks to the Author):

Summary

Early childhood exposure to antibiotics is associated with increased risk of immune diseases such as asthma, allergies and inflammatory bowel diseases, and a hypothesis is that these pathologies are the results of antibiotic-altered intestinal microbiota [Blaser & Falkow, Nature Reviews Microbiology (2009)]. While previous studies showed that massive antibiotic exposure can perturb the immunological development in mice, the authors here study the role of a single pulse antibiotic treatment (PAT) course on the intestinal microbiota and on the host developing immune system.

The presented results show that: (1) a single PAT course in early childhood is enough to produce durable alterations of intestinal microbiota and of immune system (T-cell populations and secretory IgA expressions), (2) a PAT perturbed intestinal microbiota is sufficient to transfer delayed secretory IgA expressions, and (3) early childhood exposure to antibiotics has lasting and transferable effect on microbial community network topology.

Pros.

Overall, the study is well explained and well written. Also, I don't see major methodological flaws in the experiments.

In most part, the obtained results confirm already hypothesized / measured effects of early

childhood exposure to antibiotics, with the key result that even a single PAT course can induce long lasting impairments of host immunity.

I also like the analysis of microbial association networks, although there are issues with how randomized algorithms are used (see point 4 below).

Cons.

Comment/Question 1. The study is based on a small number of mice (the smallest group having only 3 mice). While this must be balanced with the cost of the experiments (gene expression, capturing meta-genomes), this still reduces the confidence in the reported results, which is problematic given the potential importance of the presented results.

Response R2-1: Thank you for this comment. In most experiments, we had 5 or more mice/group. Please note that we were limited due to the price of some of these high-throughput technologies (e.g. Nanostring and RNAseq), however we did see consistent results across experiments, which increases the confidence in our findings. This point now is included (page 18 paragraph 1).

Comment/Question 2. The paper should position itself with respect to other recent papers about antibiotic (early) use and its consequences on microbiome [e.g., Gustafsson et al., *Journal of Immunology* (2016)]; are the results consistent? While the study mentions the differences between control and antibiotic-altered intestinal microbiota in terms of increased/decreased abundance of specific microbe species, it somehow fails at explaining if the changes are expected or not (e.g., is the species known to be associated with a diseases, or maybe its abundance also changed in other similar studies). There is only one small paragraph with one citation (page 14) which quickly discuss such relevance, which should be extended, in particular for the keystone taxa, which the authors suggest to be necessary and sufficient to explain the immunological changes.

Response R2-2: Thank you for this comment. We now cite the Gustafsson abstract (ref # 48), as suggested, and also cite papers that deal with microbiome changes after antibiotic exposures. We also further elaborate in the Discussion, the relevance of the specific microbial species that are altered with respect to phenotype and how this compares to current literature (page 15, paragraph 3, page 16 paragraph 2)

Comment/Question 3. The authors should also better explain how their results could be translated to the human case; the usage of mouse for the study of human gut microbiota has already been questioned (e.g., Nguyen et al., *Disease Models and Mechanisms*, 2015), and changes that last more than 40 days may be seen as long term changes for mice, but not necessarily for human.

Response R2-3: Thank you for comment. The conjunction of the Korpela study of antibiotic-exposed children in Finland showing prolonged effects on the microbiota from macrolides lasting months¹, and our prior PAT studies^{2,3} show similar microbial phenotypes and consistently demonstrates potent effects of macrolides. Although mouse studies cannot be a proxy to understanding the human microbiome, murine models can provide the necessary

foundations for subsequent human studies. In the Discussion, we now point to the difficulties of exactly translating mouse studies into human phenotypes, and now cite the Nguyen study (page 18, paragraph 3-19, paragraph 1; reference 49).

Comment/Question 4. In the analysis of microbial association networks, the authors use multidimensional scaling (MDS) to embed networks obtained from different conditions in 2D space according to their pairwise distances (computed using GCD), and to observe that in the 2D space, there are linear separations between networks from different conditions. Given that MDS is usually a randomized process, this experiment should be repeated at least 10 times and the results should be average over different runs to account for randomness. Similarly, the authors measured the susceptibility of these networks to random node deletions, without mentioning the percentage of deleted nodes, or the numbers of independent runs that they performed.

Response R2-4: While MDS depends on random initializations, with only five data points (networks), for the 2-dimensional embedding presented in **Figure 5**, the stress is numerically zero. This indicates we would achieve identical, global, convergence for any repetition with a random seed. For network robustness measured across random node deletions, the percentage of deleted nodes was varied between 0-100% (x-axis **Supplementary Figure S5, a-c**). We repeated the random node removal experiment N=30 times and report the average (standard error was within the thickness of the lines and hence not shown for better visibility). We now include this number in the Methods section (page 26, paragraph 3- page 27, paragraph 1).

Reviewer #3 (Remarks to the Author):

This manuscript describes the effects of a single and moderate course of antibiotics on the gut microbiota and immune development in mouse. It is shown that the effects of antibiotics treatment in early life leads to changes in microbiota and immunity that are qualitatively somewhat different and more long-lasting than the effect of a similar antibiotics treatment in adults. Comparisons to germ-free control mice and fecal transplants are used to demonstrate that the effect of antibiotics on immune development are persistent, transferable, and mediated by the microbiome. Here the paper proposes a mechanism where the antibiotics affect the species ecological network topology, making it more fragile and less resilient. The experimental design, data, statistical analysis and argumentation are robust. The manuscript is written in a very clear and concise manner; the results section is clear yet somewhat technical considering general readership.

* Major claims

This paper makes the following major claims (in a mouse model):

1) A single pulsed antibiotic course in early life has significant impact on host immunity.

This was shown to lead to a decrease in T-cell populations, intestinal lymphocytes and secretory IgA expression serum/fecal/intestinal, and expression of genes related to immunity (the effects were also greater with 3 repeated antibiotics courses compared to a single course). Changes in

immune response were not observed in germ-free controls, indicating a critical role of the microbiome in mediating these effects. Here it should be noted that the observed immunity variables were readily low in GF mice and were not further affected by antibiotics. Perturbations in the microbial community were both necessary and sufficient for observed immune response, which was further established by showing that the immune development was delayed in a fecal transplant experiment in germ-free mice who received microbiota from the antibiotics treated mice.

Comment/Question 1: Here it should be noted that the observed immunity variables were readily low in GF mice and were not further affected by antibiotics.

Response 3-1: Thank you for the comment. We have now added this point in the Results section. (Page 7, paragraph 3)

2) Antibiotics perturbation led to particular changes in the topology of the species co-occurrence network.

This is shown by demonstrating that the species network in the antibiotics treated mice is more fragile; fragility is indicated by quantifying changes in network centrality and betweenness following species removal. The effects of species removal are more remarkable in the antibiotics treated mice and the network topology changes in a way that indicates increasing fragility.

3) The antibiotics induced changes in early life microbiome and host immune phenotypes are long-lasting, persistent, and transferable to a new host.

It is shown that the transferred microbiota was as such sufficient to convey a delayed immune responses up to 11 weeks after the transfer to a new host. The gene expression changes and changes in the immune response were age-dependent and more remarkable in the pups compared to adult mice, and the community recovery was faster in adult mice than in the pups.

* Novelty and interest

- Earlier mouse studies focusing on early life immunity development have investigated the effect large antibiotics doses. The novelty in the present study is that it investigates the effects of a smaller moderate course, which is said to mimic early life antibiotics in humans and hence expected to provide useful hypotheses of the role of antibiotics in human health. This is of wide interest as similar well-controlled human studies are more challenging or impossible to conduct for obvious experimental and ethical reasons. The study design and results provide new experimental evidence that supports the earlier (and appropriately cited) hypothesis that microbiome mediates the effects of antibiotics on host immunity.

- The idea that antibiotics impact species networks is not new as such (one relevant reference that could be added is PMID:20440275). However, the targeted analysis that provide details on these effects following a moderate early life antibiotics course is new, albeit limited in scope as it focuses on taxonomic co-occurrence networks rather than the actual molecular networks and temporal dynamics.

Comment/Question 2: The idea that antibiotics impact species networks is not new as such (one relevant reference that could be added is PMID:20440275).

Response R3-2: Thank you for the comment. We have included the relevant reference as suggested, and integrated the Discussion to include the expectation of an antibiotic effect (page 17, paragraph 1, page 18 paragraph 3).

- New experimental evidence is also presented for the claim that the changes in microbiome are long-lasting, persistent and transferable, even after a moderate course of antibiotics, and that changes in the microbiome are critical for the immune development. The role of Akkermansia in iGA expression and immunity is specifically discussed in this context. The recent review by Derrien et al on the Akkermansia and host immunity (PMID:26875998) is relevant here and could be cited.

Overall, appropriate references are provided throughout the paper.

Comment/Question 3: The recent review by Derrien et al on the Akkermansia and host immunity (PMID:26875998) is relevant here and could be cited.

Response R3-3: Thank you for the comment. We now cite and discuss the recent review by Derrien et al. (page 16 paragraph 3, and new reference 29).

* Quality of evidence

- The observations are systematic and robust, and different parts of the analysis are very well complementary. The analysis has central importance for a better understanding of the long-term health implications of early life antibiotics use (effects on host immunity), and in particular its microbiome mediated nature (established by comparisons to germ-free mice and fecal transplants), including an analysis of the timing (early life vs. adults) and repetitiveness (1 vs 3 courses) of the antibiotics treatment. As such, the study helps to establish the experimental basis for studying how antibiotics impact microbial ecological networks, gene expression and immune response with potentially long-term functional consequences. As expected, the antibiotics effects also tend to be stronger in the group that received multiple treatments. Overall, the study design and experiments provide robust complementary evidence to support the main conclusions.

- The authors suggest that the antibiotics might affect microbiome structure by affecting taxonomic network topology via highly connected keystone species. Interestingly, the identity of the identified keystone species seems to be robust to antibiotics dosing and microbiome transfer. Moreover, the analysis indicates that the antibiotics perturbed networks are more fragile (more sensitive to species removal), and that the network structure is an essential element in the transferable nature of the persisting effects of antibiotics. The network analysis is limited, however, as it is primarily based on taxonomic co-occurrence networks and lacking analysis of the network dynamics within individuals over time. Measures of network topology are only indirect measures of fragility. Therefore, the fragility could be shown experimentally but this would require study designs that are out of scope for this work. This limitation could be stated in

the discussion, however. Alternatively, the indicated fragility measures could be a side product of switching community types (PMID:26866806). This could be cited and discussed. As such, the experimental evidence on microbiome fragility is preliminary and does not cover molecular functional mechanisms or temporal dynamics, although the former is being discussed. Gathering further experimental evidence on how antibiotics treatment affects the network dynamics and molecular function would be interesting but does not seem feasible for the scope of this paper but the limitations could be stated in the discussion.

Comments/Questions 4: Therefore, the fragility could be shown experimentally but this would require study designs that are out of scope for this work. This limitation could be stated in the discussion, however. Alternatively, the indicated fragility measures could be a side product of switching community types (PMID:26866806). This could be cited and discussed. As such, the experimental evidence on microbiome fragility is preliminary and does not cover molecular functional mechanisms or temporal dynamics, although the former is being discussed. Gathering further experimental evidence on how antibiotics treatment affects the network dynamics and molecular function would be interesting but does not seem feasible for the scope of this paper but the limitations could be stated in the discussion.

Response R3-4: Thank you for the comment. The reviewer is quite correct to mention that our fragility measures are several steps removed from experimental ecology, dynamics or mechanisms. This limitation has been added to the Discussion section (substituting ‘suggest’ rather than ‘show’ and ‘may be’ rather than ‘is’. Now written as: The dynamics of the ecological and physiologic changes suggest the importance of network stability, which may be especially at risk in early life (page 19, paragraph 2). The reviewer also proposes an intriguing idea: measures of network fragility may be tied to [switching between] community types. However, we believe that this discussion is outside the scope of this study since our observed networks are not based on dynamic models, nor do we explore generative, topological models. Since it is not possible to know if network fragility is a cause or consequence of community type switching – such speculations would likely not contribute to the discussion.

Comment/Question 5: The follow-up times are 40-77 days. To me this seems an intermediate rather than long-term follow-up. Some clarification might help.

Response R3-5: Thank you for the comment. Previous transfer experiments surveyed microbial communities for up to 4 weeks (ref # 38). In the studies that we now report, we extend the observation time to 11 weeks to assess the characteristics of the colonization by the transferred microbiota and whether the microbiota remains distinct (antibiotic-perturbed vs control). We now clarify the duration of the transfer study and that we were able to use intermediate time-points as well as the end-point at sacrifice (page 8 paragraph 3- page 9 paragraph 1 and page 17 paragraph 2).

Comment/Question 6: Recently, it was proposed that exposure to diverse microbiomes in the living environment is essential for restoring microbial balance after microbiota is depleted. I think there was a very recent journal paper which I can't find now. But there is at least a recent abstract on this topic <https://academic.oup.com/ecco-jcc/article/2961708/P768> - since delayed

recovery of the microbiome is a key component in this paper, could the delay possibly be due to the fact that the living environment does not provide the normal microbiota (does not accommodate normal mice, for instance). Ideally, the study design would control also for this factor but it may not be feasible in the scope of the current study. This factor could have a remarkable effect on the results and should be discussed in the manuscript. Was this controlled?

Response R3-6: Thank you for the comment. Antibiotic (PAT) treated mice remained in the same vivarium after treatment (which only was for 5 days), using mice in different cages as controls. Cages were changed weekly; thus both the PAT and Control groups were exposed to the same background in the vivarium; the environment would have been the same for the control and the PAT groups. We now include the issue of restoration of the microbiota and room assignment in the Methods section (page 21 paragraph 2- page 22, paragraph 1).

* Impact on thinking in the field

- This work provides a robust and useful benchmark for further studies aiming to elucidate the mechanistic link between early life antibiotics, microbiome perturbation, and later life health issues. It covers relevant aspects of the single antibiotics course in early life, including the effect on microbiome composition and co-occurrence network topology, functional implications related to host immune development, and the demonstration that these effects are age-dependent, persistent over time and transferable to a new host.

- The study design and results provide essential guidance for further and more detailed analysis on the underlying mechanisms. The main contribution of the paper is to gather solid experimental evidence for topical hypotheses that are central for contemporary host-microbiome research, rather than in the development of new methods or concepts.

- The main readers will be researchers who study host-associated microbiomes, including the human microbiome, and in particular those focusing on the microbiome early-life development and function, which a currently very active subfield within microbiome research. Also other fields of microbial ecology where microbiome manipulation by antibiotics or by other means is an active research topic may benefit from the findings.

Comment/Question 7: The results section is clear but rather technical; a schematic figure of the experimental setup and observations would be useful; the study design has various aspects with varying time scales and experimental groups. While reading the manuscript, it was difficult to keep track on all these different aspects, and a figure summarizing them all in a single schematic figure (aided at a general readership) would be helpful.

Response R3-7: Thank you for the comment. We now include a Supplemental Figure with Study design that addresses the issues raised by the reviewer (new Supplementary **Figure 6**).

* Statistical analyses

- Overall, the statistical analyses have been performed rigorously, including appropriate controls, replication, and sample size. The conclusions are supported by the data. I have some comments:

Comment/Question 8: The inter- and intra-group divergence is defined as the mean pairwise unweighted UniFrac divergences (as described in Fig. 1 caption for instance). Although this has been sometimes used in literature to quantify group divergence, it is critical to note that the mean of all pairwise divergences within a group is sensitive to the group size, and mean is also a biased measure of the group divergence since the various pairwise dissimilarities are not independent (A vs B; B vs C; A vs C) and these problems grow rapidly with the group size. One solution would be to select a smaller subset of independent random pairs from the group, or comparing the divergences from the group mean within each group between the two groups. At the minimum, the group sizes in the comparison should be equal to avoid bias coming from the group size. Can you comment if these considerations were taken into account when comparing inter- and intra group divergences?

Response R3-8: We now address the issue of independence between group divergences (in the Methods section, page 25, paragraph 3- page 26), by subsampling the pairwise comparisons as suggested by the reviewer, and have updated the figure accordingly. In short, we generated inter-group UniFrac distances averaged over independently-drawn sample pairs (subsampling without replacement and replicated 999 times).

Comment/Question 9: DESeq2 and ANOVA are used to identify significant differences between groups in RNA profiling data. This seems suboptimal as recent studies (PMID:26028277 and PMID:24699258) have demonstrated high false discovery rates for the t-test (this will generalize to anova), and for DESeq2 (PMID:28253908) in this context compared to other methods that take better into account the nature of sequencing-based microbiome profiling data. The differences in the performance of the alternative methods were so notable that this issue should be addressed. At the minimum, the possible effects of this choice on the analysis results should be discussed and, ideally, the results would be updated or at least confirmed with an alternative method that that has been demonstrated to work well in the recent comparative reviews. Many of those improved algorithms are readily available in R/Python and the calculation are straightforward, so this should be feasible.

Response R3-10: Thank you for the comment. We agree with the reviewer about the suboptimal approaches of using DESeq2 on microbial abundance data. As there are many methods for normalization and statistical testing, we sought to determine differentially abundant taxa using LEfSe (PMID:21702898), which implements a non-parametric Kruskal-Wallis rank-sum test and Linear Discriminate Analysis (LDA). Our host ileal transcriptomic (RNA-sequencing) was performed using DESeq2, which has been found to be a reliable method for transcriptomic studies (PMID:26539333). We have updated the text to avoid any confusion between the analysis of 16S-V4 rRNA from microbial communities and RNA from the host tissue, and have included text about the limitations of our statistical methods (in the Statistical approaches section; Page 22, paragraph 2).

* Reproducibility; data and code availability

- Methods description: Sufficient methodological details for reproducing the experiments are

provided.

Comment/Question 11: Data availability: data has been made only partially available; the RNAseq and the 16S rRNA metagenomic data are being made available via standard public repositories. However, I could not check these data sets as they are not open at the time of this review. Therefore, I can't confirm whether sufficient sample metadata is available to repeat the analyses. Data on the iGA profiling seem to be missing as well, unless it is included in the cited ArrayExpress/QIITA repositories that I can't access.

Response R3-11: See below, we now have deposited all of the data as requested (see below R3-12)

Comment/Question 12: Code availability: references to custom scripts and open source tools are made but no code has been provided. As the source code of the statistical analyses is missing, many potentially relevant technical details will remain hidden. I would strongly suggest providing the full analysis source code available as supplementary methods to accompany the paper online. The code should be sufficiently well documented so that the manuscript results can be derived from the source data (which is also provided). Ideally, the code could be used to directly reproduce the reported results based on the released data. This effort should be moderate and feasible to implement.

Response R3-12: Thank you for the comment. Along with the main text of the manuscript, and to maximize transparency and reproducibility of our study, the sequence data we generated have been deposited in multiple publicly available databases.

The RNAseq data have been deposited in the ArrayExpress database with the accession code: E-MTAB-5101 (<http://www.ebi.ac.uk/arrayexpress/experiments/E-MTAB-5101/>). (<http://www.ebi.ac.uk/arrayexpress/experiments/E-MTAB-5101/>). (Please see the Cover letter for reviewer login information)

Our 16S-V4 data can be found within the QIITA repository under the Study ID 10527 (<https://qiita.ucsd.edu/study/description/10527>).

The ileal Nanostring data have been deposited in NCBI's Gene Expression Omnibus (Edgar *et al.*, 2002) and are accessible through GEO SuperSeries accession number: GSE98292 (<https://www.ncbi.nlm.nih.gov/geo/query/acc.cgi?acc=GSE98292>). (Please see the Cover letter for reviewer login information)

More importantly, we generated reproducible Jupyter and RMarkdown notebooks that contain the commands and data used to generate the main figures found within the manuscript (<https://github.com/blaser-lab/Paper-Ruiz-2017>). These notebooks, combined with the publicly available datasets, can be used to fully and completely reproduce the findings described in the manuscript.

- Availability of other materials: I can't comment on the availability of reagents or other wetlab materials as that is out of my speciality.

Reviewer #4 (Remarks to the Author):

In this manuscript, Ruiz and colleagues address the question of how early-life antibiotic exposure affects the long-term composition of the microbiome and the impact it has on intestinal immune homeostasis. Given the continually emerging data that demonstrates a role for the microbiome in shaping immune responsiveness, and the continuing need for antibiotic courses to treat childhood infections, this reductionist study performed in mouse models provides a useful framework for gaining mechanistic understanding of how these interventions could shape immunity. Despite the excitement associated with these studies, there are a few concerns about the study design and the overall novelty of these findings.

Comment/Question 1: In the Introduction, the authors cite a few papers that have used long-term exposure to broad-spectrum cocktails of antibiotics to interrogate the role of the resident microbiota in immune development and function. The authors rightly state that these doses are not representative of how antibiotics are administered to people, especially children. However, they overlook publications that have addressed how early-life exposure to single antibiotics can influence the outcome of disease in models of allergic asthma (SL Russell & BB Finlay, PMID: 25145536 and PMID: 23333861). The antibiotic dosing used in these manuscripts is different to what is being investigated here, but should be used to contextualize the findings reported here.

Response R4-1: Thank you for the comment. We now cite the Russell papers in the context of a model of disease, asthma, and as an example of the relevance of our findings (page 18 paragraph 3-19 paragraph 1)

Comment/Question 2: The antibiotic treatment course used here is very similar to what was reported in a previous publication from the lab of the corresponding author. This increases confidence that the results are consistent (decreased Bacterioides, increased Enterobacteriaceae), but there are some notable differences as well (increased Akkermansia). Akkermansia has mucus-degrading function, and germ-free mice or mice treated with the antibiotic cocktail, have a diminished mucus barrier. It is curious that Akkermansia is increased in this setting – have the authors determined whether the mucus lining is affected by their antibiotic treatment protocol?

Response R4-2: Thank you for the comment. We have not measured mucin levels in this study, and we will mention this limitation in the Discussion section (Page 16, paragraph 2). The literature about Akkermansia shows varied relationships with pathogenic phenotypes—sometimes associated, sometimes protective. We now comment about the increased Akkermansia in this study, which confirms our prior reports.^{2,3} This consistency in three independent studies provides evidence that the findings are robust.

Comment/Question 3: In Figure 2, the authors characterize immune changes associated with antibiotic treatment. They report that mice that were exposed to PAT have no change in total CD4⁺ T cell numbers, or the frequency of Treg and IFN γ ⁺ cells, but a decrease in the frequency of Th17 cells. It stands to reason that there could be a relative increase in the frequency of another CD4⁺ Th lineage – have the authors investigated whether there is a skew toward Th2

polarization in the PAT-treated mice? This would be relevant to the observation that germ-free or antibiotic-cocktail treated mice are more Th2-like and could be informative for our understanding of how early life exposure to a diverse microbiota shapes immune polarization.

Response R4-3: Thank you for this comment. Although we did not measure Th2 cytokine levels, we did examine expression of cytokines and other genes associated with Th2 responses, including *Il4*, *Il4R*, *Il5*, *Il9*, *Il13* and *Gata3* in the nanostring analyses. We did not find any significant deviation in expression of these genes. Although prior studies suggest that antibiotics may promote Th2 polarization, the timing of the antibiotic doses and type of antibiotic class used may substantially affect the outcome. We now highlight these points in the Discussion section (page 15, paragraph 2).

Comment/Question 4: In Figure 2, the authors determine that fecal levels of IgA are diminished by PAT1 or PAT3. Given the recent publication showing microbe-dependent degradation of IgA (Moon, Baldrige & Virgin PMID: PMC4425643) it is important that the data be presented in a way that accounts for this. Are the dams that are shown in the right-hand panels the same dams that gave rise to the pups in the left-hand panels? To provide an extra level of confidence that PAT results in decreased sIgA in pups, the authors could monitor sIgA in pups born to an untreated dam, then monitor sIgA in a second litter of pups that come from that same dam + PAT.

Response R4-4: Thank you for this comment. We now have measured sIgA levels in the dams before breeding and at gestation day 16; this includes dams who later received PAT or not (Controls). We now show that the dams had comparable levels of fecal sIgA (see new Supplemental Table S4). Although we did not collect samples from their progeny before antibiotic treatment since they were too young for collections, based on Moon et al⁴, we can deduce that the sIgA levels also are comparable in the pups before PAT treatment. We include this point in the text (page 6 paragraph 1) now clarify Figure 1 and its Legend to show that the sIgA levels in the pups are the offspring of the Dams on the left.

Comment/Question 5: It is unclear why the authors chose to focus their analysis on ileal tissue given that the majority of the bacterial biomass is in the colon. These assays could be reported on to demonstrate whether there are tissue-specific differences or if there is global disruption of intestinal immune homeostasis.

Response R4-5: The colon is packed with bacteria and the expression of many genes is constitutive, even in germ-free mice⁵; In contrast, host gene expression in the ileum is much more susceptible to the effect of microbial colonization than in the colon⁵. We had anticipated that there would be substantial impact based our prior analyses of ileum with PAT (in NOD mice)³. In fact, in the first PAT experiment we report here (Figure 2), we found significant differential effects in the expression of 148 (27.1%) of the 547 host immune genes studied between PAT and control. We now provide this rationale for the studies we conducted (page 5 paragraph 2).

Comment/Question 6: On page 7, the authors state that “importantly, the profiles for PAT-exposed SPF mice clustered with the GF mice, and not with the SPF controls (Figure 3g), even

40 days after the antibiotic exposure, indicating a persistent effect.” This statement should be discussed and compared to their previous PAT paper where they concluded, "Although the PAT groups largely converged with control over time, differences never fully resolved." (Ref 13 in submitted manuscript)

Response R4-6: We thank the reviewer for his/her careful reading of our studies. In the prior paper², we show that even >90 days after the antibiotic exposure ended, the PAT group had not recovered the richness or evenness (alpha diversity) of the control groups (Figures 3,5 of that paper). The current results are similar in the types of differences, although we did not carry out the follow-up as long (up to day 77 in the transfer study shown in Figure 4). We show the changes persisting more than 40 days after exposure ended (Figure 1; page 4 paragraph – page 5, paragraph 1) in the single course experiment, and for 77 days in the transfer experiment (Figure 4; page 9, paragraph 2- page10).

Comment/Question 7: For the PAT-treatment of germ-free mice, it would be worth providing the rationale that macrolides have been proposed to immunomodulatory in the Results section, rather than waiting to introduce the concept in the Discussion.

Response R4-7: Thank you, we now have made this change (page 7 paragraph 2).

Comment/Question 8: Following immune characterization, the authors perform comprehensive analysis of the microbiota, ultimately concluding “Taken together, these results allow the prediction that perturbation of these keystone taxa profoundly impacts the stability of the overall community; especially since their keystone roles were conserved even after transfer into GF mice.” While the amount of computational analysis done to get to this place is appreciated, it begs functional analysis. Can the authors test this provocative hypothesis and relate it to the immunophenotypes characterized earlier?

Response R4-8: The reviewer correctly points out the limitations on keystone assessment in the network description. As such, we have modified the above statement to “; especially since their keystone *network attributes* were conserved even after transfer into GF mice”, and we have removed mention of keystone species. (We refer to potential keystone taxa, and keystone network attributes to better indicate the uncertainty associated with these statements rather than using the term ‘keystone species’ (page 14 paragraph 2). The relationship between keystone species and immunological roles will need to be addressed in a future study.

References for the Response letter

- 1 Korpela, K. *et al.* Intestinal microbiome is related to lifetime antibiotic use in Finnish pre-school children. *Nat Commun* **7**, 10410, doi:10.1038/ncomms10410 (2016).
- 2 Nobel, Y. R. *et al.* Metabolic and metagenomic outcomes from early-life pulsed antibiotic treatment. *Nat Commun* **6**, 7486, doi:10.1038/ncomms8486 (2015).
- 3 Livanos AE *et al.* Antibiotic-mediated gut microbiome perturbation accelerates development of type 1 diabetes in mice. *Nature Microbiology* doi:10.1038/nmicrobiol.2016.140. (2016).
- 4 Moon, C. *et al.* Vertically transmitted faecal IgA levels determine extra-chromosomal phenotypic variation. *Nature* **521**, 90-93, doi:10.1038/nature14139 (2015).

- 5 Larsson, E. *et al.* Analysis of gut microbial regulation of host gene expression along the length of the gut and regulation of gut microbial ecology through MyD88. *Gut* **61**, 1124-1131, doi:10.1136/gutjnl-2011-301104 (2012).

REVIEWERS' COMMENTS:

Reviewer #1 (Remarks to the Author):

All reviewers' comments were addressed - the only issue remains relates to the impact of the antibiotic administration on weight and other physiological properties. Here the authors refer to upcoming studies but do not give any details or provide results.

Reviewer #2 (Remarks to the Author):

The authors adequately addressed the comments.

Reviewer #3 (Remarks to the Author):

The authors have adequately addressed my comments and discussed on the limitations has been added.

Reviewer #4 (Remarks to the Author):

Thank you to the authors for your efforts revising this manuscript. My comments have been sufficiently addressed.

Response to Reviewers (NCOMMS-17-03244A)

Reviewer 1

Comment: All reviewers' comments were addressed - the only issue remains relates to the impact of the antibiotic administration on weight and other physiological properties. Here the authors refer to upcoming studies but do not give any details or provide results.

Response: As we discussed previously, we are working on a separate manuscript that details the effects of the PAT antibiotic administration on body weight and body composition. This manuscript already is quite long and complex, and we believe that we cannot do justice to the metabolic effects without an extensive addition to the current manuscript. As such, we respectfully request to not include this area in the current manuscript.

Reviewers 2, 3, 4

No comments were made.